# (Be Cautious!) Bio-Foundation Models Are Not Yet Robust to Biological Plausible Perturbations and ML Transformations

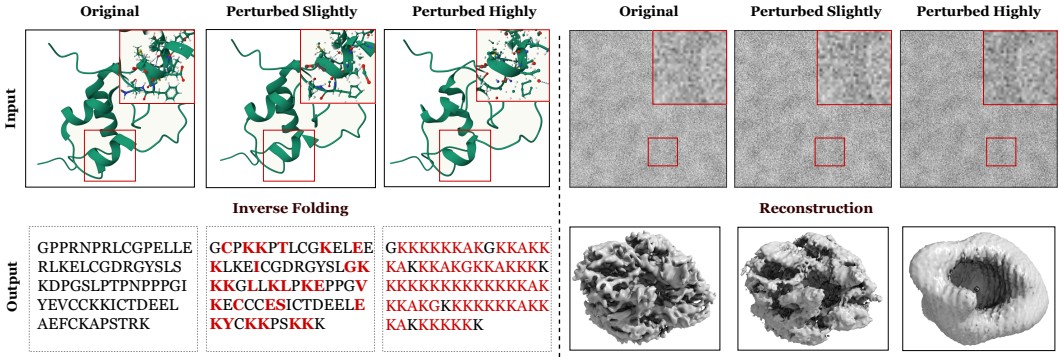

Figure 1: Illustration of biologically plausible perturbations and their downstream effects across structural and imaging modalities. **Left:** Protein structure perturbations applied to atomic coordinates and annotations. The upper panel shows perturbed protein backbones, while the lower panel depicts the corresponding outputs from inverse folding (sequence recovery) after perturbation. **Right:** Cryo-EM image perturbations simulating experimental artifacts and noise. The upper panel shows corrupted cryo-EM particle images, and the lower panel presents reconstructed 3D densities obtained from these perturbed inputs.

## Abstract

Biological Foundation Models (Bio-FMs) have demonstrated remarkable success across diverse biomedical domains, enabling advances in drug discovery, protein design, and molecular analysis. However, the robustness of Bio-FMs remains underexplored, particularly in terms of the unique risks and perturbations they may encounter in real-world deployment and how these challenges impact their utility. In this work, we characterize the robustness of Bio-FMs from both biology and machine learning (ML) perspectives, and we observe that Bio-FMs are not yet robust to biological data curation and ML transformations. Specifically, (i) from the biological data curation perspective, we design biologically plausible perturbations that mimic corruptions commonly observed in biological experiments, and assess their impact on Bio-FMs; (ii) from the ML perspective, we probe how data transformations, preprocessing, and embedding affect model performance. We systematically evaluate state-of-the-art Bio-FMs on a spectrum of protein-related downstream tasks, spanning protein design, generation, function prediction, cryo-EM reconstruction, and structure classification, over structure, sequence, and image modalities. Our results reveal that most Bio-FMs are vulnerable to both ML transformations and biological perturbations; however, cryo-EM reconstruction models (e.g., CryoDRGN) exhibit a surprising robustness, which maintains stability even under worst-case adversarial scenarios. Notably, we also find that subtle biological perturbations, which are often imperceptible to current measurement tools, yet induce severe discrepancies in Bio-FM outputs, leading to critical failures. Our work highlights underappreciated vulnerabilities and provides a new perspective for evaluating and improving the trustworthiness of Bio-FMs.

# 1 INTRODUCTION

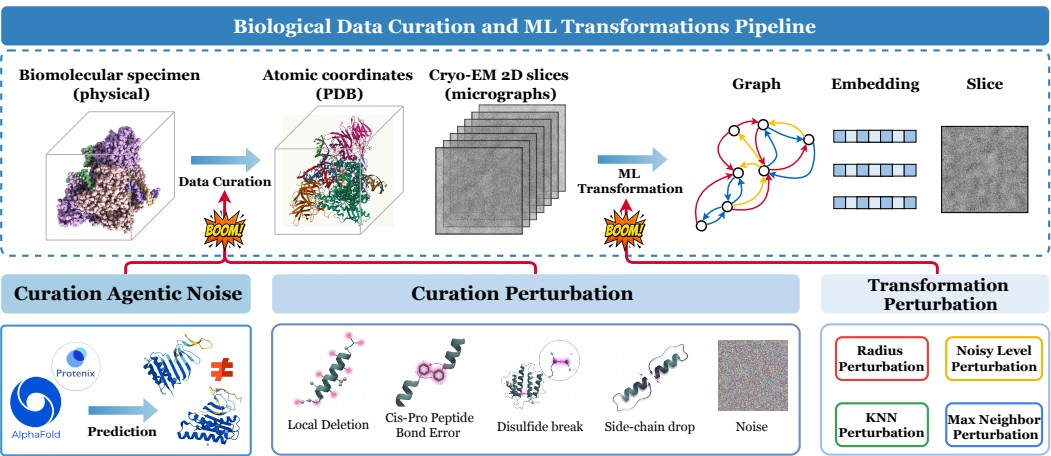

Figure 2: The biologically plausible data perturbation and ML transformations pipeline. The biologically plausible data perturbation includes geometric and coordinate-level perturbations and annotation and format-level perturbations. The ML transformations consider data and representation transformations perturbations.

The recent development of biological foundation models (Bio-FMs) has enabled inspiring success in deciphering biological molecules, ranging from individual proteins (Jumper et al., 2021b), single-cell RNA sequences (Theodoris et al., 2023) to large molecular complexes (Zhou et al., 2022; Baek et al., 2024; Guo et al., 2024; Lu et al., 2022a; Corso et al., 2022). This rapidly growing community has significantly accelerated the discovery and design of novel molecules, substantially advancing real-world biomedical applications such as therapeutic development, drug discovery, and vaccine design (Zhang et al., 2025; Sharma et al., 2022).

However, despite these remarkable breakthroughs, the robustness of Bio-FMs in real-world deployment remains largely unexplored. Most recently, a preliminary study (Lyu et al., 2025) reveals that both AlphaFold2 (Jumper et al., 2021b) and AlphaFold3 (Abramson et al., 2024b) exhibit systematic flaws in reproducing biomolecular energetics, raising concerns about the reliability of these leading Bio-FMs. At the same time, several correspondences (Bloomfield et al., 2024; Wang et al., 2025) have drawn global attention to the broader safety issues surrounding Bio-FMs.

In this work, we aim to call attention to the robustness issues of Bio-FMs and to ensure their dependable use by presenting a principled and systematic study of their robustness from both biological and machine learning (ML) perspectives. Specifically, we raise the following critical questions that remain to be answered: (1) *What kinds of factors in real-world deployment may influence the reliability of these biological models?* (2) *Under what conditions are they most likely to fail?* (3) *To what extent are their predictions and practical utilities affected by real-world perturbations or interference?*

To investigate these questions, we characterize and highlight key differences between Bio-FMs and general foundation models in real-world deployment, as illustrated in Figure 2, focusing on two key aspects: biologically plausible perturbation and ML transformations. Any changes, noise, or perturbations within each procedure can introduce robustness issues for Bio-FMs, which are often overlooked or assumed to be ideal during their development. Therefore, in this work, we aim to provide a systematic study from these two perspectives and specifically propose **biologically plausible perturbations** and **machine learning transformations** to analyze their detailed effects on model performance. Considering the wide range of Bio-FMs, we focus our study on leading protein-based Bio-FMs across structure, sequence, and image modalities (*i.e.*, cryo-EM images used for 3D molecular reconstruction). In biologically plausible perturbations, we introduce noise commonly encountered when curating folding data from models such as AlphaFold; geometric-level perturbations; coordinate-level perturbations; format-level perturbations, and corruption noise observed during biological data acquisition for structure, sequence, and image modalities. From the machine learning

perspective, machine learning transformations include data and representation transformations performed within Bio-FMs, including internal parameters such as the radius and $k$NN parameters used to construct graph representations within the model.

In summary, we benchmark 2,128 experiments across 11 state-of-the-art Bio-FMs, spanning 7 datasets and 4 categories of downstream tasks. As a result, we find that even subtle perturbations to the input can induce large changes in the predictions of state-of-the-art models, as illustrated in Figure 1. Specifically, on a spectrum of protein-related downstream tasks, including protein design, generation, structure classification, function prediction and cryo-EM reconstruction, we reveal that most Bio-FMs are vulnerable to both ML transformations and biological perturbations at different severity levels, while we find that cryo-EM reconstruction models (e.g., CryoDRGN) exhibit a surprising robustness, which maintains stability even under worst-case adversarial attacks. Additionally, we also find that subtle biological perturbations, which are often imperceptible to current measurement tools, yet induce severe discrepancies in Bio-FM outputs, leading to critical failures.

In a nutshell, our contributions can be summarized as follows: (1) To the best of our knowledge, we are the first to present a systematic and comprehensive study of biological robustness from both biological and machine learning perspectives. (2) We identify key robustness challenges of Bio-FMs and introduce biologically plausible perturbations and machine learning transformations to evaluate and benchmark the robustness of leading Bio-FMs. (3) We investigate a broad spectrum of protein-based Bio-FMs and applications across sequence, structure, and image modalities to provide a comprehensive robustness analysis. (4) Through extensive evaluations on seven datasets spanning different modalities, we reveal the vulnerability of current Bio-FMs under varying degrees of perturbation and demonstrate their adverse impact on downstream applications.

## 2 RELATED WORK

### 2.1 BIOLOGICAL FOUNDATION MODELS

Recently, the development of biological foundation models, drawing inspiration from the success of large language models, have significantly accelerated biological molecular analysis and design. Early efforts, such as ProGen (Madani et al., 2023), relied solely on autoregressive pretraining over protein sequences. However, the generated sequences often lacked connections with the corresponding 3D structures. To enable more effective representation learning and structure-aware design, many recent works explicitly incorporate 3-D structural or geometric information alongside sequences. For instance, GearNet (Zhang et al., 2023c) introduces a geometry-aware relational graph neural network that represents proteins as graphs with residue-level nodes and connected by diverse edge types, pretrained via multiview contrastive learning. Similarly, ProNet (Wang et al., 2023a) employs a 3D graph network for structure-aware protein representation, but with a hierarchical design to capture multi-level structural information. SaProt (Su et al., 2024) extends the sequence modeling paradigm by augmenting the vocabulary with structure tokens, enabling the training of a structure-aware protein language model. ProSST (Li et al., 2024) further advances this direction by quantizing protein structures into discrete tokens through a structure-encoding module, and then applying disentangled attention in a Transformer to model interactions between residue tokens and structure tokens. In parallel, the AlphaFold family Jumper et al. (2021a); Abramson et al. (2024a); Baek et al. (2024), with its transformer blocks over MSA columns and pair matrices, has achieved unprecedented accuracy in protein structure prediction and provided representations that strongly benefit downstream protein design. The ESM family (Bjerregaard et al., 2025; Hsu et al., 2022; Lin et al., 2022) complements these advances by scaling protein pretraining to billions of sequences and embedding multiple data modalities jointly. Besides sequence and structure modalities, the emergence of cryo-electron microscopy (cryo-EM) enables high-resolution visualization of biomolecules in near-native states and has encouraged the development of machine learning models (Zhong et al., 2021a;b; Huang et al., 2024b; Qu et al., 2025b; Liu et al., 2023; Herreros et al., 2025; Lu et al., 2022b; Punjani et al., 2017; Qu et al., 2025a) for automatic reconstruction of 3D molecular structures from image inputs for structural analysis.

### 2.2 SECURITY AND ROBUSTNESS IN FOUNDATION MODELS

With the rapid development of powerful foundation models, concerns about their security in real-world applications have grown significantly (Das et al., 2025; Yu et al., 2025; Ma et al., 2025; Huang

et al., 2024a; Zhang et al., 2024a). For instance, large language models (LLMs) have been shown to be vulnerable to attacks such as prompt injection and distribution shifts, which can trigger harmful or misleading outputs (Perez & Ribeiro, 2022; Crothers et al., 2023). Likewise, vision (Kirillov et al., 2023) and vision–language foundation models (Shayegani et al., 2023) are highly susceptible to adversarial perturbations. For instance, Segment Anything Model (SAM)(Kirillov et al., 2023) can be compromised by adversarial examples, resulting in a severe degradation of segmentation accuracy(Long et al., 2025). For biological foundation models, robustness issues are only beginning to be explored, yet they are particularly critical given the close connection to high-stakes biological applications. Jha et al. (2021) show that structure predictions from RoseTTAFold (Baek et al., 2021) can change drastically under very small sequence perturbations. Similarly, Yuan et al. (2023) investigate adversarial sequence mutations against the AlphaFold2 model. More recently, SafeGenes (Zhan & Moore, 2025) demonstrates that genomic foundation models, including ESM (Meier et al., 2021), suffer substantial performance degradation under targeted soft-prompt attacks. In parallel, SafeProtein (Fan et al., 2025) introduces robustness benchmarks for protein foundation models, calling for greater attention to this direction.

## 3 BIO-FM ROBUSTNESS FROM ML AND BIOLOGY PERSPECTIVES

### 3.1 PRELIMINARY

Biological foundation models (Bio-FMs) are large-scale pretrained models that learn universal representations from vast biological data, such as sequences, structures, and images, and serve as adaptable backbones for diverse downstream biomedical tasks (Guo et al., 2025). In Table 1, we present the taxonomy of the Bio-FMs involved in this work, with their core characteristics and task domains. We conduct a comprehensive investigation of more than 10 state-of-the-art Bio-FMs spanning protein design, sequence generation, function prediction, structural classification, and cryo-EM reconstruction, over extensive datasets and input modalities. In Appendix A, we present the detailed description of each Bio-FM and the conducted tasks. We provide detailed illustrations of the perturbation scope for each model in Appendix B.1.

### 3.2 MOTIVATIONS AND CHALLENGES

Recent biological studies highlight the unreliable behaviors of Bio-FMs, raising concerns about their reliability in critical biomedical applications. For instance, researchers have recently uncovered systematic failure patterns of AlphaFold3 (Baek et al., 2024), even when tasked with predicting protein structures that are close to its training distribution. Such findings underscore a fundamental question: *What are the underlying sources of vulnerability in Bio-FMs?*

**General FMs vs. Bio-FMs** General FMs usually operate on data that is human-generated and largely symbolic (text and images), where perturbations mostly arise from ML-side transformations, such as data corruption, pre-processing, and embeddings. In contrast, Bio-FMs operate on biological manifolds that are inherently physical and biochemical (e.g., protein sequences, 3D structures, cryo-EM images). These are not just "curated data points" but representations of natural objects with fragile physical constraints. Moreover, biological data are prone to experimental noise and sample preparation artifacts (e.g., noisy cryo-EM reconstruction errors, sequencing misreads, protein misfolding states). Unlike text or image corpora, these errors are not always human-detectable or correctable. Tiny biological perturbations (e.g., a single amino acid mutation, thermal fluctuation in cryo-EM) may be invisible to standard tools but can catastrophically alter Bio-FM outputs. This makes biological curation risks fundamentally different, as they introduce "silent" vulnerabilities invisible to standard ML robustness pipelines. Therefore, we argue that the robustness failures of Bio-FMs can stem from both inference-time **ML transformations** and **biologically plausible perturbation**.

### 3.3 BIO-FM PERTURBATIONS FROM ML AND BIOLOGY PERSPECTIVES

In this paper, we investigate the robustness of Bio-FMs from two complementary angles: the ML side and the biologically plausible perturbation side: (i) from the **ML perspective**, we examine how internal data and representation transformations (e.g., *protein graph embeddings*, *tokenization*, and

Table 1: The taxonomy of protein-related biological downstream tasks and biological foundational models (or tools) involved in this work. "seq." stands for "sequence". "ML" and "Bio." stand for perturbations from ML and biological perspectives, respectively.

| Downstream Tasks | Model | Dataset | Metric | Input Modality | Perturbation Scope |
|---|---|---|---|---|---|
| Function or Structure Prediction | GearNet (Zhang et al., 2023c) | Enzyme Commission (EC) | AUPRC | Structure | ML, Bio. |
| | ESM-GearNet (Zhang et al., 2023a) | Gene Ontology (GO) | F1 | Structure | ML, Bio. |
| | ESM-1 (Meier et al., 2021) | ProtFunc | Accuracy | Sequence + Structure | Bio. |
| | ProNet (Wang et al., 2023b) | HomologyTAPE | | Structure | ML, Bio. |
| Sequence Generation | ESM-3 (Hayes et al., 2025) | PInvBench | | Structure | Bio. |
| | ProteinMPNN (Dauparas et al., 2022) | (mpnn validation) | Recovery Rate | Structure | ML, Bio. |
| | ESM-IF1 (Hsu et al., 2022) | | | Structure | Bio. |
| Protein 3D Reconstruction | CryoDRGN (Zhong et al., 2021a) | RAG1–RAG2 complex | Fourier Shell Correlation | cryo-EM | ML, Bio. |
| | CryoNeRF (Qu et al., 2025b) | (EMPIAR-10049) | (FSC) | cryo-EM | ML, Bio. |
| Protein Fitness Prediction | SaProt (Su et al., 2024) | | Spearman | Structure | Bio. |
| | ESM-3 (Hayes et al., 2025) | ProteinGym | AUC | Structure | Bio. |
| | ESM-IF1 (Hsu et al., 2022) | (DMS-substitution, | Recall | Sequence + Structure | Bio. |
| | S3F (Zhang et al., 2024b) | DMS-indels) | | Sequence + Structure | ML, Bio. |
| | ProtinMPNN (Dauparas et al., 2022) | | | Sequence + Structure | ML, Bio. |

*preprocessing*, shape the stability and robustness of Bio-FMs (Section 4); (ii) from the **biological perspective**, we study how naturally occurring and frequently observed corruptions during data curation (e.g., *amino acid coordinate shifts*, *geometric distortions*, and *sequence mutations*), impact Bio-FM performance (Section 5). These analyses provide a dual view of robustness that reflects both the computational transformations inherent to Bio-FMs and the biological perturbations rooted in real-world data collection.

## 4 ML TRANSFORMATIONS REMAIN A THREAT TO BIO-FMS' ROBUSTNESS

### 4.1 SETUP: ML TRANSFORMATIONS INSIDE BIO-FMS

ML-side perturbations are defined as inference-time transformations that occur within the internal pipelines of Bio-FMs, such as preprocessing, embedding, and tokenization schemes. For example, when processing protein structural information, Bio-FMs often encode structures into graphs by connecting residues as nodes with edges determined by spatial proximity. In this step, ML considerations, such as the number of neighbors or the cutoff radius used to capture spatial relations, can significantly alter the resulting graph representation and the model's behavior. Inference-time perturbations test the reliability of Bio-FMs under slight data shifts and can uncover deeper aspects of their robustness in real-world deployment. Notably, these transformations are independent of the biological data curation process, assuming that the biological data has already been generated and fixed in advance. Evaluating ML-side perturbations is thus essential to disentangle robustness issues arising from computational design choices and enables a clearer understanding of how Bio-FMs fail or succeed under different modeling assumptions.

Since protein is one of the most popular research objects in Bio-FMs, we mainly consider the ML perturbations that happen in protein structure modeling, such as protein graph construction. In Appendix B.1, we provide the detailed perturbation strategy, including the transformations considered in each Bio-FM, as well as the perturbation configurations. In summary, we perturb the spatial relationships and density distributions in protein graph modeling across multiple Bio-FMs, with various strengths.

**Similarity Measurement.** As in prior robustness studies, defining how to measure the distance between original and perturbed data is critical, particularly when auditing the feasibility, utility, and broader practical implications of robustness analysis in real-world applications. Here, we quantify perturbation strength using *graph similarity* metrics, including spectral distance, Frobenius norm, and Jaccard Similarity over edges. In Appendix B.2, we present the detailed calculation procedures of these measurements. By default, we utilize the Jaccard Similarity over edges as the similarity measurement.

### 4.2 BIO-FMS ARE NOT YET ROBUST TO ML TRANSFORMATIONS

**Probing the Robust Boundary of Bio-FMs.** To provide a comprehensive understanding of the robustness of Bio-FMs against ML transformations, in Figure 3 we probe the robustness boundary of S3F, ESM-GearNet, GearNet, and ProteinMPNN across various benchmarks. Specifically, each point in Figure 3 represents a perturbation caused by a different ML transformation, where the x-

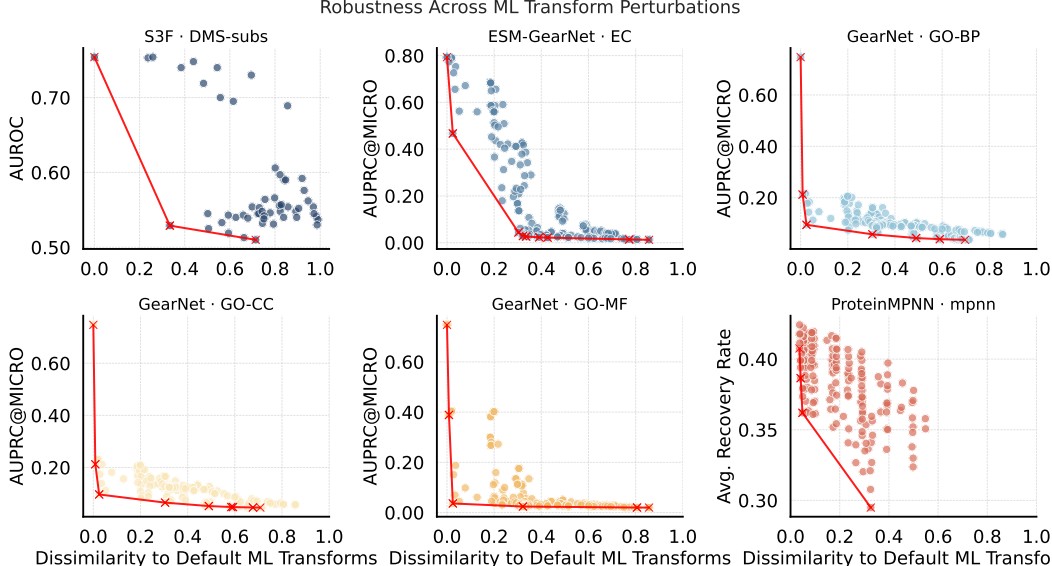

Figure 3: Probing the robust boundary of Bio-FMs in terms of ML transformations. We observe that tiny perturbations (measured by graph Jaccard similarity) result in significant performance drops in various Bio-FMs. This suggests that existing Bio-FMs are not robust to ML transformations and require further consideration in real-world deployment.

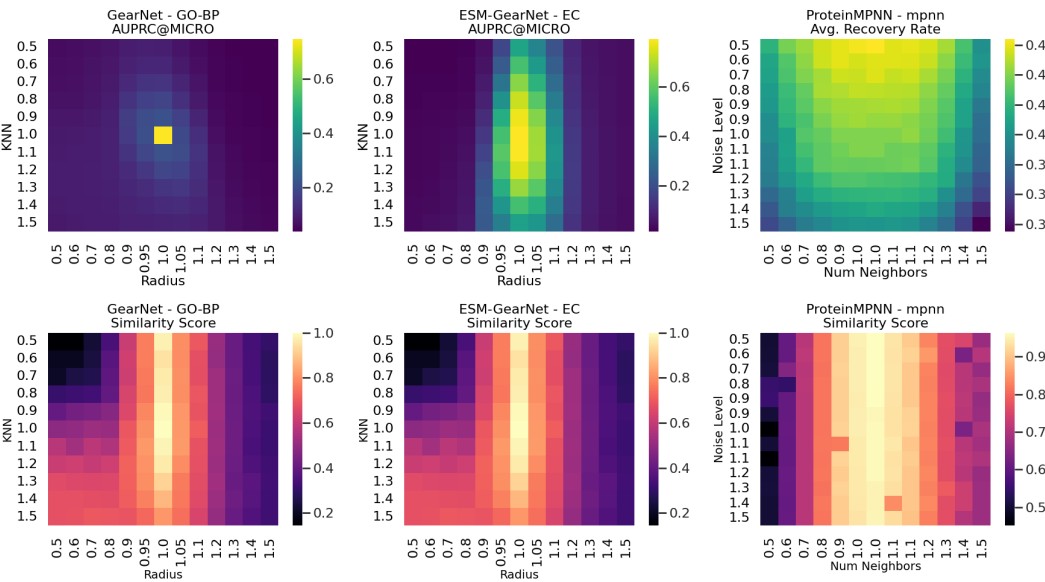

Figure 4: The performance and similarity heatmap over various perturbation sources. It is shown that GearNet is extremely vulnerable: a slight increase in radius during protein graph construction in the testing time will significantly hurt performance.

axis measures the dissimilarity (*i.e.*, 1− similarity) relative to the default transformation, and the y-axis shows the corresponding performance on the evaluation benchmarks. We then plot the lower-envelope curve (red line) to indicate the worst-case boundary under these perturbations. Despite varying degrees of sensitivity, all Bio-FMs exhibit a drastic performance drop within a very small range of perturbation, as measured by dissimilarity. For instance, GearNet, the least robust model, drops from 0.7 to 0.1 AUPRC@MICRO when the dissimilarity is as low as 1%.

Table 2: We investigate the impact of biologically plausible perturbations on cryo-EM data for the Protein 3D Reconstruction task. The quality of the reconstructions is evaluated using Fourier Shell Correlation (FSC), where a lower FSC value indicates better accuracy. We report FSC results under different levels of perturbation severity to assess the robustness of different methods.

| Severity | Gaussian Blur | | Rotation | | Translation | | PGD Attack |
|---|---|---|---|---|---|---|---|
| | CryoDGRN | CryoNeRF | CryoDGRN | CryoNeRF | CryoDGRN | CryoNeRF | CryoDGRN |
| 1 | 3.503 | 3.667 | 3.502 | 3.663 | 3.502 | 3.712 | 3.502 |
| 3 | 3.503 | 3.754 | 3.736 | 4.195 | 7.205 | 7.688 | 3.501 |
| 5 | 8.612 | 9.968 | 4.574 | 6.899 | 64.663 | 66.755 | 3.502 |

**Tiny Perturbations Result in Significant Performance Drops.** The robustness boundary motivates a deeper diagnosis of model behavior under specific ML transformations. As shown in Figure 4, the top row presents performance variations as different ML transformation parameters change, where the coordinate axes represent the variation scales of each parameter, while the bottom row depicts the corresponding similarity changes. For GearNet and ESM-GearNet, we vary the radius and the $k$ value (default $k = 10$) in $k$NN when constructing the multi-relational GNN. GearNet is extremely sensitive to both parameters: even tiny changes in either the $k$ value or the radius can lead to complete model failure. This severity is further highlighted by the similarity plot on the bottom: the constructed graphs across different $k$ values maintain high similarity, yet the performance drops sharply with only a small

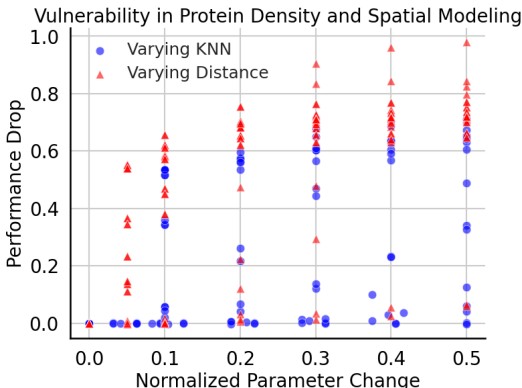

Figure 5: *The Vulneralbility of Density and Spatial Modeling.* We show the performance drop due to changes in normalized parameters with two modeling strategies: *density modeling*, denoted as $k$-NN, and *spatial modeling*, denoted as Distance.

change in $k$. In contrast, ESM-GearNet exhibits greater robustness to variations in $k$, maintaining its performance over a relatively wider $k$ range, while remaining sensitive to small changes in the radius. The similarity plot is identical to that of GearNet, as both models construct input graphs in the same way but differ in the algorithms used for processing inputs and making predictions. Besides, for ProteinMPNN, we vary number of neighbors and noise level parameters when constructing the graph representations, with results shown in the third column. ProteinMPNN demonstrates more robustness to perturbations in graph representation construction. Note that the model applies noise by defaultk, therefore decreasing the noise level leads to a slight increase in performance, whereas varying the number of neighbors exerts a stronger influence on model performance.

**Vulnerability of Density and Spatial Modeling.** To further investigate the vulnerability of Bio-FMs to perturbations in graph construction, we examine two commonly used modeling strategies: *density modeling*, exemplified by $k$NN, and *spatial modeling*, where graph edges are established based on atom distance thresholds. The results are shown in Figure 5, where performance degradation is plotted against normalized parameter changes. Across different levels of parameter variation, we find that current Bio-FMs are more vulnerable to spatial modeling, which exhibiting consistently larger performance drop under the same degree of change.

## 5 BIOLOGICALLY PLAUSIBLE PERTURBATION POSES INHERENT CHALLENGES TO BIO-FMS' ROBUSTNESS

### 5.1 SETUP: BIOLOGICAL PLAUSIBLE PERTURBATIONS

To systematically evaluate robustness to real-world data issues, we develop a comprehensive suite of biologist-driven, biologically plausible perturbations spanning both protein structures and cryo-EM images. These perturbations are engineered to mimic common errors and artifacts that arise during experimental data curation. For protein structures, our perturbations are categorized into two

classes. (1) Geometric and coordinate-level perturbations that directly alter the physical representation of the molecule. Examples include applying Gaussian noise to atomic coordinates to simulate thermal fluctuations, introducing local deletions of residue segments to mimic unresolved loops or regions of poor electron density. (2) Annotation and format-level perturbations that introduce errors into the protein structures file's metadata and structure. Examples include scrambling B-factor and occupancy values, which encode atomic mobility and confidence, and removing or breaking critical records that define chain boundaries and chemical connectivity.

For the cryo-EM imaging modality, we introduce a set of image perturbations designed to simulate experimental artifacts such as low signal-to-noise ratios, defocus effects, and sample heterogeneity. Specifically, we apply various noise models (Gaussian, shot, impulse, and speckle noise) (McMullan et al., 2016; Li et al., 2013; Rice et al., 2018), image quality degradations (Gaussian blur and low contrast)(Zhang, 2016; Glaeser, 2013), and geometric transformations (rotation, translation, and elastic transforms) (Afanasyev et al., 2015; Zheng et al., 2017; Scheres, 2012). These corruptions represent a range of realistic scenarios, from ice contamination to particle misalignment. In addition to these natural corruptions, we assess worst-case vulnerability by employing a Projected Gradient Descent (PGD)(Madry et al., 2017) method to generate adversarial perturbations.

Finally, we assess agentic pipeline risk to model error propagation in multi-stage workflows. In this setup, a 3D structure generated by a prediction model (e.g., AlphaFold) is fed to downstream Bio-FMs. This process reveals how inherent prediction uncertainties from an upstream model can cascade and create vulnerabilities in subsequent ones. A detailed description of each perturbation method is available in the Appendix C.

## 5.2 How Biologically Plausible Data Perturbation Hurts Bio-FMs?

**Biologically Plausible Perturbations.** Similar to our study of ML transformations, we begin by examining the robustness boundary under perturbations introduced during the biological curation process. As shown in Figure 6, each point represents a randomly applied biologically plausible perturbation, where we compute the dissimilarity between graphs constructed with input before and after the perturbation and report the corresponding benchmark performance. Even with small dissimilarity changes, the worst-case performance of each Bio-FM decreases drastically, indicating that robustness issues are severe when Bio-FMs are exposed to perturbations arising from real-world data curation.

**Different Bio-FMs Respond Differently to Specific Biological Perturbations.** Furthermore, we investigate two biologically plausible perturbations that frequently arise during biological data curation: ❶ *coordinate perturbation*, where coordinate values are fluctuated by adding Gaussian noise, and ❷ *rename perturbation*, where residues are incorrectly labeled during sequence formatting. As shown in Figure 8, we examine the behavior of ESM3 and ProNet under both perturbations. We observe

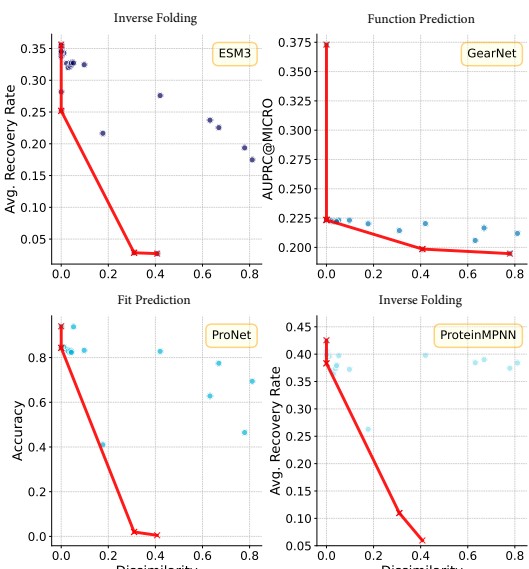

Figure 6: *Biological Perturbation Robustness Boundary.* We demonstrate the dissimilarity (i.e., $1 - similarity$) between graphs constructed from inputs before and after perturbation, plotted on the x-axis, along with the corresponding model's task performance on the y-axis.

moderate robustness for both models at low perturbation levels, but their performance collapses when the perturbation severity exceeds four. Under the rename perturbation, ESM3 demonstrates poor robustness, likely due to its heavy reliance on sequence-based training, whereas ProNet remains comparatively stable owing to its structure-focused design.

**Bio-FM Uncertainty Risks Agentic Pipeline.** Bio-FMs are deployed in agentic pipelines for therapeutic design, such as combining ESM3 or ProteinMPNN with AlphaFold3 for rapid antibody

development. However, our results reveal a critical robustness challenge in Bio-Agentic systems: Bio-FMs may transmit incorrect uncertainty signals to downstream tasks, creating significant risks. In Figure 7, we conduct antibody design experiments where ProteinMPNN generates antibody candidates, AlphaFold3 predicts their structures, and Rosetta (Alford et al., 2017) evaluates their free energy. While AlphaFold3 reports highly consistent ptm/iptm scores, the corresponding Rosetta energy calculations show large variance. This disparity underscores a robustness risk: stable Bio-FM confidence does not guarantee stable downstream behavior. Subtly encoded uncertainties—undetected by AlphaFold3's self-reported metrics—can propagate into downstream evaluations, leading to substantial shifts in conclusions about antibody fitness.

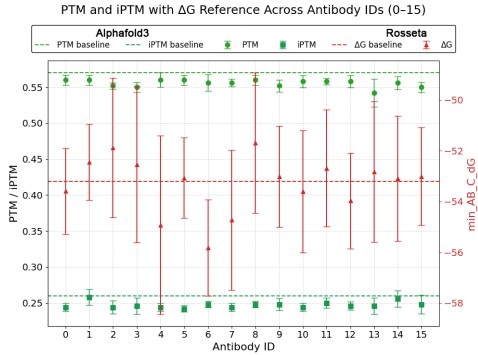

Figure 7: Antibody design in an agentic system. Alphafold3 provides high-confidence marker (ptm/iptm) yet result in huge variance in downstream tasks (Rosetta Free Energy).

**Cryo-EM Reconstruction Models Are Robust, Even Worst-Case.** As shown in Table 2, the Cryo-EM reconstruction model is robust against biologically plausible perturbation. Specifically, (1) the FSC remains below 0.5 under perturbations like Gaussian Blue, Rotation with severity less than and equal to 3.

This indicates that these perturbations do not lead the model to confuse noise with a valid signal, except in cases of extremely high noise, which are implausible in real-world scenarios. (2) For translation perturbations, the FSC exceeds 0.5 under large perturbation, *i.e.*, when severity is greater than and equal to 3. (3) In the case of worst-case perturbations, such as the PGD attack, our model remains stable across different severity levels, specifically: $6/192$ for level 1 severity, $12/192$ for level 2 severity, and $12/192$ for level 3 severity. We attribute such superior robustness of Cryo-EM models (e.g., CryoDRGN) compared to structure/sequence models (e.g., GearNet, ProNet) to three key factors: Information Aggregation, Training Objectives, and Input Continuity. Please refer to Appendix D for more discussion.

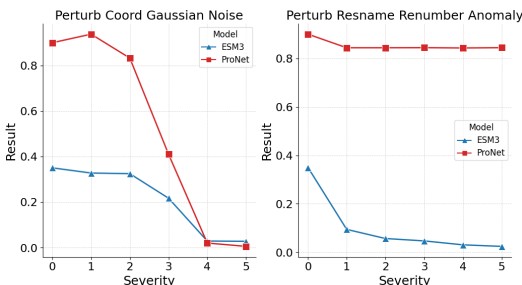

Figure 8: *Biological Perturbation on Different Bio-FMs.* We show two types of biologically plausible perturbations: (1) *coordinate perturbation* (left), and (2) *rename perturbation* (right). We plot the performance change for different levels of severity of ESM2 and ProNet.

**Non-FM tool Robustness.** We take enzyme function prediction Yu et al. (2023) as an example and use BLAST as a non-FM conventional tool, as shown in Table 3. BLAST transfers the EC annotation of the closest homologous sequence identified through high-scoring alignments. We show that BLAST is not being affected in 8 out of 12 perturbations. This is because BLAST takes a sequence as input and matches the enzyme function via sequence similarity from external databases. Naturally, it immunizes spatial distance perturbations such as the Gaussian coordinate perturbation. While Bio-FMs models the spatial protein structures and significantly suffers from Gaussian coordinate perturbation (e.g., GearNet drops from 0.76 to 0.65). Even for the rest 4 perturbations, BLAST demonstrates strong resilience, e.g., BLAST only drops 3-4% accuracy on average and drops only 7% at most in the worst scenario. This shows that although Bio-FMs are highly capable on many tasks, they also exhibit greater vulnerability compared to traditional non-FM bio tools. We will expand our discussion and include additional results, such as broader comparisons between Bio-FMs and non-FM tools on the same tasks, to further illustrate this point in our manuscript.

## 6 CONCLUSION

In this paper, we propose a systematic and comprehensive analysis of biological robustness from both biological and machine learning perspectives. This novel approach highlights the importance

Table 3: Non-FM tool robustness analysis with BLAST and enzyme function prediction. It is shown that non-FM biological tools are more robust to biological perturbations compared to Bio-FMs.

| Dataset | Halogenase | | Multi | | New | |
|---|---|---|---|---|---|---|
| Ori BLAST | 0.66 | 0.66 | 0.18 | 0.18 | 0.52 | 0.52 |
| **Bio. Perturb.** | **Severity1** | **Severity3** | **Severity1** | **Severity3** | **Severity1** | **Severity3** |
| Gaussian Coordinate Noise | 0.66 | 0.59 | 0.18 | 0.18 | 0.52 | 0.51 |
| Local Residue Deletion | 0.62 | 0.62 | 0.18 | 0.15 | 0.51 | 0.50 |
| Sidechain Atom Drop | 0.66 | 0.66 | 0.18 | 0.18 | 0.52 | 0.52 |
| Disulfide Bond Breakage | 0.66 | 0.66 | 0.18 | 0.18 | 0.52 | 0.52 |
| Cis-Peptide Bond Error | 0.66 | 0.66 | 0.18 | 0.18 | 0.52 | 0.52 |
| Local Geometric Distortion | 0.66 | 0.66 | 0.18 | 0.18 | 0.52 | 0.52 |
| B-Factor and Occupancy Scrambling | 0.66 | 0.66 | 0.18 | 0.18 | 0.52 | 0.52 |
| Atom Name/Element Misalignment | 0.66 | 0.66 | 0.18 | 0.18 | 0.52 | 0.52 |
| Residue Name and Numbering Anomalies | 0.59 | 0.66 | 0.18 | 0.18 | 0.52 | 0.52 |
| Header and Terminator Record Corruption | 0.66 | 0.66 | 0.18 | 0.18 | 0.52 | 0.52 |
| CONECT Record Loss | 0.66 | 0.66 | 0.18 | 0.18 | 0.52 | 0.52 |

of robustness for bio-foundation models. We identify two key perturbations of bio-foundation model robustness: biologically plausible perturbations and machine learning transformations. These two types of perturbation affect the robustness of bio-foundation models both during data curation and model training, covering the model from development to application. Specifically, our study explores robustness across diverse modalities, including sequence, structure, and image. This systematic analysis provides a comprehensive overview of robustness for bio-foundation models. Our results indicate that developers should pay attention to these previously ignored robustness issues, which are critical for the safe utilization of biological models.

**Limitations.** While our work provides a systematic benchmark for Bio-FM robustness, we recognize several promising directions for future research. Our analysis could be extended to an even broader range of models and tasks as the field rapidly evolves. Furthermore, connecting our in silico findings with experimental validation remains an important next step to fully understand the real-world impact of these vulnerabilities. Finally, delving deeper into the mechanistic underpinnings of why certain models exhibit robustness offers a valuable path toward designing the next generation of more reliable and trustworthy Bio-FMs.

## ETHICS STATEMENT

We follow the ICLR Code of Ethics. Our study involves no private, sensitive, or personally identifiable information. We anticipate no ethical issues nor harmful societal impacts arising from this work.

## REPRODUCIBILITY STATEMENT

Reproducibility is a core aim of our study. All experimental datasets are publicly available standard benchmarks. The main paper and appendix provide complete details of the training procedures, model architectures, and evaluation metrics. Upon acceptance, we will release the full codebase—including preprocessing, training, and evaluation scripts—along with configuration files and documentation to enable exact replication of our results. Random seeds and hyperparameters will also be provided to further ensure reproducibility.

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

## A    BIOLOGICAL FOUNDATION MODELS AND DOWNSTREAM TASKS

### A.1    BIOLOGICAL DOWNSTREAM TASKS

**Function or Structure Prediction.**    This category covers predicting molecular function (e.g., EC and GO annotations, interface/ligand binding) and inferring 3D structure or structural proxies from available inputs (Jumper et al., 2021b). In practice, Bio-FMs provide transferable sequence/structure embeddings that are consumed by lightweight heads for classification or regression, or they directly produce structural outputs. These tasks probe whether pretraining captures biophysical constraints, evolutionary regularities, and fold-level inductive biases that generalize across families. They are foundational for proteome-scale annotation, mechanism-of-action studies, and for bootstrapping downstream design pipelines that depend on reliable structure/function priors.

**Sequence Generation.**    Here the goal is *de novo* protein design: proposing amino-acid sequences that are likely to fold, remain stable, and achieve target properties (e.g., binding, catalysis, trafficking) (Madani et al., 2023). Models operate either purely in sequence space (autoregressive/Masked LM sampling with constraints) or condition on structure/backbone contexts to steer designs. Typical evaluation includes sequence recovery under fixed backbones, in silico stability or binding proxies, and wet-lab validation when available. By efficiently traversing an astronomically large sequence space, Bio-FMs accelerate discovery beyond natural diversity while enabling multi-objective optimization.

**Protein 3D Reconstruction.**    Given many noisy 2D cryo-EM projections, the task is to infer high-resolution 3D densities and, increasingly, the continuous landscape of conformational states. Modern deep generative approaches learn mappings from images to volumes and latent variables describing heterogeneity, improving resolution and handling flexibility/partial occupancy (Zhong et al., 2021a). Accurate reconstructions are essential for visualizing assemblies, understanding allostery, and providing structure priors for docking and design. They also stress-test robustness, since small imaging artifacts or alignment errors can cascade into markedly different volumetric solutions.

**Protein Fitness Prediction.**    Fitness prediction estimates the effect of mutations (substitutions and indels) on activity, stability, binding, or organismal viability—i.e., learning the fitness landscape. Bio-FMs score variants using sequence likelihoods, structure-aware encoders, or multi-scale surface/geometry features, and are evaluated on deep mutational scanning benchmarks (Meier et al., 2021). Reliable fitness models guide directed evolution, variant prioritization, and safety analysis by highlighting deleterious or gain-of-function changes. They also serve as a stringent test of whether embeddings encode causal, not merely correlational, signals linking sequence, structure, and function.

### A.2    BIOLOGICAL FOUNDATION MODELS

**ProNet.**    A hierarchical protein representation learner based on complete 3D graph networks that captures residue-, substructure-, and protein-level signals. It ingests protein structures as graphs (residue or atom nodes with edges from chemical connectivity and spatial proximity) to compute expressive embeddings. Typical uses include function classification (EC/GO), interface/binding-site prediction, stability/property regression, and family/homology classification with whole-graph features (Wang et al., 2022).

**GearNet.**    A multi-relational GNN for proteins with message passing over sequence-adjacent edges, spatial neighbors, and $k$NN graphs to couple primary sequence and tertiary geometry. It operates on residue-level 3D graphs augmented with geometric and physicochemical features to produce node- or graph-level representations. Applications include function prediction, active-site annotation, and structure-aware property prediction, providing strong structure-conditioned baselines (Zhang et al., 2022).

**ESM-GearNet.**    A hybrid architecture that fuses ESM language-model embeddings with a Gear-Net structural encoder to jointly leverage evolutionary and geometric information. It takes amino-acid sequences for the ESM component and 3D structure/graphs for GearNet, aligning the modali-

ties into a unified embedding. The combined representation improves EC/GO classification, binding/property prediction, and homology transfer over single-modality encoders (Zhang et al., 2023b).

**ProteinMPNN.** A protein designing model that design sequences for a given protein backbone structure. It outperforms traditional physically-based methods in terms of native sequence recovery and computational efficiency, and successfully rescues previously failed designs across a wide range of protein design challenges. The output sequences leading to higher AlphaFold prediction accuracy, and demonstrate improved experimental expression, thermostability, and correct assembly in diverse applications (Dauparas et al., 2022).

**ESM-1.** First-generation Evolutionary Scale Modeling transformers trained on massive protein sequence corpora to learn universal language representations of proteins. Inputs are linear amino-acid sequences, from which residue and sequence embeddings are derived via masked-language-modeling objectives. Resulting features support classification tasks, secondary/contact proxies, remote homology detection, and zero-shot mutation scoring for fitness ranking via language-model likelihoods (Meier et al., 2021).

**ESM-3.** A multi-track, multi-task Bio-FM that couples sequence modeling with structural/geometric signals and iterative generative refinement. It can consume sequences together with structure tokens/coordinates or geometry-aware attention biases to form joint representations. Capabilities span joint sequence–structure reasoning, sequence generation/design, and structure-aware annotation, including conditional design under backbone or functional constraints (Hsu et al., 2022).

**ESM-IF (inverse folding).** A structure-to-sequence model trained to generate or rank sequences compatible with a given backbone, effectively solving the reverse of folding. It takes 3D backbones or coordinate traces (e.g., $C_\alpha$ or backbone frames), optionally with side-chain context, and outputs per-position amino-acid distributions or full sequences. Common uses include design under fixed folds and compatibility scoring for mutations and scaffolds.

**S2F.** A sequence–structure fitness framework that integrates PLM-derived sequence embeddings with geometric encoders (e.g., GNNs/GVPs) to model mutation effects (Zhang et al., 2024b). It consumes both the amino-acid sequence and a 3D structure or predicted backbone to produce multimodal representations. These features are trained for fitness prediction on DMS and variant panels, typically generalizing better than sequence-only scoring.

**S3F.** An extension of S2F that adds an explicit protein-surface representation (mesh or point cloud) to capture pockets, interfaces, and local topology (Zhang et al., 2024b). Inputs comprise sequence, 3D structure, and surface geometry/features, which are encoded at multiple scales. The resulting embeddings achieve state-of-the-art performance on fitness prediction and variant ranking, particularly for interface-mediated phenotypes.

**SaProt.** A structure-aware protein language model that augments the token vocabulary with structure-derived tokens, injecting geometric context during language modeling. It processes sequences annotated with discretized local geometry or related structural cues to produce more structure-sensitive embeddings. These embeddings improve structure/function prediction and stability/fitness classification over sequence-only PLMs on structure-dependent endpoints (Su et al., 2023).

**CryoDRGN.** A variational deep generative model for cryo-EM that maps 2D particle images into a latent space of 3D densities, capturing continuous conformational heterogeneity. It ingests particle images (with viewing parameters/poses) and decodes latent variables into volumetric densities consistent with observed projections. Outputs support 3D reconstruction and conformational landscape analysis, handling heterogeneous ensembles more naturally than single-state pipelines (Zhong et al., 2021a).

**CryoNeRF.** A neural radiance field (NeRF) formulation of cryo-EM reconstruction that learns a continuous volumetric field whose projections match measured images. Given cryo-EM images and estimated poses/orientations, it fits an implicit function over 3D coordinates to recover high-fidelity

densities. The approach extends to heterogeneous states via conditioning on latent variables and offers smooth, grid-free volumetric representations (Qu et al., 2025b).

# B ML TRANSFORMATIONS INSIDE BIO-FMS

## B.1 ML TRANSFORMATIONS

In contrast to perturbations that simulate experimental or annotation errors, this category targets the internal data processing and representation choices within the Bio-FMs, as shown in table 4. Specifically, we investigate the sensitivity of models to the hyperparameters governing the construction of protein graphs, which are fundamental data structures for many structure-aware models. These inference-time transformations probe the stability of a model with respect to its own architectural and preprocessing assumptions. The specific parameters perturbed for each model are detailed below, with ranges selected around their default values.

- **GearNet & ESM-GearNet:** These models construct protein graphs based on spatial proximity. We perturb two key hyperparameters that define the graph topology:
    - `radius`: This hyperparameter defines the cutoff distance (in Å) for connecting residues as nodes with an edge. A larger radius results in a denser graph. We perturb this value within the range of $\{5, \ldots, 15\}$ Å, where the default is 10 Å.
    - `KNN`: As an alternative to a fixed radius, this method connects each residue to its $k$ nearest neighbors based on Euclidean distance. This ensures a uniform node degree across the graph. We vary the number of neighbors $k$ across the set $\{5, \ldots, 15\}$, with a default value of 10.
- **ProNet:** This model also relies on a graph representation, and we perturb its graph construction parameters:
    - `cutoff`: Similar to GearNet's radius, this parameter sets the distance threshold for building spatial edges between residues. It is perturbed over the range $\{5, \ldots, 15\}$ Å, with a default of 10 Å.
    - `max_num_neighbors`: This parameter imposes a hard cap on the maximum number of neighbors for any given residue, thereby controlling the maximum node degree and graph density. We evaluate the model's robustness to this constraint by varying the limit from $\{16, \ldots, 48\}$, where the default is 32.
- **S3F:** This model's geometric encoder uses distance-based criteria to form edges, which we perturb as follows:
    - `min_distance`: This parameter sets a lower bound on the distance for an edge to be considered valid, effectively filtering out residue pairs that are too close. We perturb this value across $\{5, \ldots, 15\}$ Å, centered on the default of 10 Å.
    - `radius`: This parameter acts as the upper cutoff distance for connecting edges. We evaluate a range of $\{0, 4, \ldots, 32\}$ Å. The default value of 0 typically disables this filter, so our perturbations test the effect of introducing and varying this spatial constraint.
- **ProteinMPNN:** This model uses a graph-based representation to inform its sequence generation process. We perturb two key aspects of its internal mechanism:
    - `num_neighbors`: This hyperparameter controls the size of the local neighborhood (number of nearest residues) considered during the message-passing steps for predicting an amino acid at a given position. We vary this number from $\{24, \ldots, 72\}$, with a default of 48.
    - `noise_level`: The model adds Gaussian noise to atomic coordinates during training for regularization. We test the model's sensitivity to this factor at inference time by applying noise with a standard deviation varying across $\{0.1, \ldots, 0.3\}$ Å, around the training default of 0.2 Å.

The ML transformations focus on perturbations in the graph construction of protein structures. For models such as ESM-1, ESM-3, ESM-IF1, and SaProt, which do not involve graph construction,

Table 4: The ML-perspective perturbations involved in our work.

| Bio-FMs | Transformation | Explanation | Perturbation Range |
|---|---|---|---|
| GearNet | radius | Defines the cutoff distance (in Å) for connecting atoms into edges. | {5 ... 15} (default 10) |
| | KNN | Connect each residue to its k nearest neighbors (based on Euclidean 3D distance). | {5 ... 15} (default 10) |
| ProNet | cutoff | Defines distance cutoff for building spatial edges. | {5 ... 15} (default 10) |
| | max_num_neighbors | A cap on how many neighbors each residue can connect to. | {16 ... 48} (default 32) |
| ESM-GearNet | radius | Nodes within this distance are considered spatial neighbors. | {5 ... 15} (default 10) |
| | KNN | Connect each residue to its k nearest neighbors (based on Euclidean 3D distance). | {5 ... 15} (default 10) |
| S3F | min_distance | Lower bound on distances considered valid edges to filter out too-close pairs. | {5 ... 15} (default 10) |
| | radius | Upper cutoff distance for connecting edges, same as above. | {0, 4, 8, ... 32} (default 0) |
| ProteinMPNN | num_neighors | Controls how many nearest residues are considered when predicting an amino acid. | {24 ... 72} (default 48) |
| | noise_level | Adds Gaussian noise to atomic coordinates during training. | {0.1 ... 0.3} (default 0.2) |

we do not apply these ML transformation perturbations. Instead, we integrate only biologically plausible perturbations (BioPP) for these models. For models related to Protein 3D Reconstruction, the input data are images. In this case, ML transformation perturbations align with biologically plausible perturbations like Gaussian Blur, Rotation, and Translation. Additionally, we adopt the gradient attack method as a type of ML transformation. Specifically, we apply the PGD Attack to perturb Protein 3D Reconstruction tasks.

### B.2 SIMILARITY MEASUREMENT

To quantify the structural dissimilarity induced by the ML transformations on the protein graph representations, we employ a suite of metrics that capture changes at both local and global scales. These metrics measure the distance between the original graph $G = (V, E)$ and the perturbed graph $G' = (V, E')$.

- **Jaccard Similarity**: This metric provides a direct measure of edge overlap (Jaccard, 1901) and is defined as the size of the intersection of the edge sets divided by the size of their union: $|E \cap E'|/|E \cup E'|$. A value of 1 indicates identical graphs, while a value of 0 indicates no shared edges. This metric offers a straightforward and interpretable quantification of how local residue connectivity is altered by the perturbation.

- **Frobenius Distance**: Calculated on the adjacency matrices $A$ and $A'$ of the two graphs (Horn & Johnson, 2012), the Frobenius distance is defined as $\|A - A'\|_F$. This is the square root of the sum of the squared differences between the elements of the matrices. It is sensitive to the exact number of edges that differ between the two graphs, effectively measuring the magnitude of the change in the adjacency representation.

- **Spectral Distance**: This metric assesses changes in the global topological properties of the graph (Chung, 1997). It is computed as the Euclidean distance ($L_2$-norm) between the sorted vectors of eigenvalues (the spectra) derived from the graph Laplacian matrices, $L$ and $L'$. Since the spectrum of a graph encodes fundamental structural information, such as connectivity, the number of components, and the presence of bipartite structures, a small spectral distance implies that the perturbed graph maintains global properties similar to the original.

## C    BIOLOGICALLY PLAUSIBLE PERTURBATIONS DURING DATA CURATION

This appendix provides a comprehensive technical description of the biologically plausible perturbations designed and implemented for this study. These perturbations are engineered to mimic common errors, artifacts, and variations that occur during the experimental data acquisition and curation pipelines for protein structures and cryo-electron microscopy (cryo-EM) images (MRC format). Each perturbation is controlled by a severity parameter, an integer from 1 (mildest) to 5 (most severe), which maps to specific corruption parameters.

### C.1    PERTURBATIONS FOR PROTEIN STRUCTURES

Our PDB perturbations are divided into two categories: (1) those that alter the physical 3D coordinates and (2) those that corrupt the file's annotation and formatting, which can challenge parsing and interpretation by downstream models.

### C.1.1 Geometric and Coordinate-Level Perturbations

These perturbations directly modify the atomic coordinates, simulating physical and experimental uncertainties.

- **Gaussian Coordinate Noise:** This simulates thermal fluctuations and positional uncertainty inherent in experimentally determined structures (Djinovic-Carugo & Carugo, 2015; Atilgan et al., 2001). We add Gaussian noise sampled from $\mathcal{N}(0, \sigma^2)$ to the (x, y, z) coordinates of every atom. The standard deviation $\sigma$ (in Ångströms) is determined by the severity level: (0.10, 0.20, 0.40, 0.80, 1.20) for severities 1 through 5, respectively.

- **Local Residue Deletion:** This mimics unresolved loops or regions of poor electron density where a segment of the protein chain cannot be modeled (Chen et al., 2010; Leaver-Fay et al., 2011). For each chain, we delete a continuous segment of residues. The deletion is preferentially applied to the middle of the chain to better simulate loop regions. The length of the deleted segment is a fraction of the total chain length, with the fraction frac mapped from severity as: (0.02, 0.04, 0.06, 0.08, 0.12).

- **Sidechain Atom Drop:** This simulates incomplete modeling of flexible or low-resolution sidechains (Engh & Huber, 1991; Vendruscolo et al., 2002). For each residue, with a given probability prob, we remove all of its sidechain atoms. The backbone atoms (N, CA, C, O) and the CB atom are preserved to maintain the basic residue structure. The probability prob for dropping a sidechain is: (0.05, 0.10, 0.18, 0.25, 0.35).

- **Disulfide Bond Breakage:** This simulates errors in modeling covalent disulfide bonds or changes in the local redox environment (Jabs et al., 1999; Tozzini, 2005). We first identify potential disulfide bonds by finding pairs of Cysteine SG atoms within a 2.3 Å distance. For each identified pair, with a probability prob, we break the bond by deleting one of the two SG atoms. The breakage probability prob is: (0.3, 0.5, 0.7, 0.85, 1.0).

- **Cis-Peptide Bond Error:** This introduces a geometrically incorrect peptide bond conformation, which is a known, albeit rare, modeling error (Karplus & Kuriyan, 2005; Tirion, 1996). We specifically target the peptide bond preceding a Proline residue (X-Pro), which is naturally found in a *trans* conformation ($> 99\%$ of cases). We simulate a forced transition towards a *cis* conformation by rotating the Proline residue around the C(i)-N(i+1) peptide bond axis. The rotation angle rot_deg is chosen to approach the $180°$ flip required for a full *trans*-to-*cis* switch: (60, 90, 120, 150, 170)°.

- **Local Geometric Distortion:** This simulates localized strain or subtle inaccuracies in bond lengths and angles within a residue (Carugo & Carugo, 2005). A fraction cover of residues in each chain are randomly selected. For each selected residue, we apply a minor affine transformation to its atomic coordinates. The transformation consists of an anisotropic scaling and a slight shear, centered on the residue's geometric center. The scaling factor for each axis is drawn from $1 \pm$ scale_span. The parameters are mapped from severity as:
  - cover: (0.05, 0.10, 0.15, 0.22, 0.30)
  - scale_span: (0.02, 0.04, 0.06, 0.08, 0.12)

### C.1.2 Annotation and Format-Level Perturbations

These text-based perturbations introduce errors into the PDB file's metadata and structural records, challenging the robustness of data parsers.

- **B-Factor and Occupancy Scrambling:** This corrupts the B-factor and occupancy columns, which encode atomic mobility and conformational confidence. Depending on severity (Kleywegt & Jones, 1996), we apply different schemes:
  - *Severity 1-2:* B-factors are shuffled across all atoms, and occupancies are randomized by sampling from $\mathcal{N}(0.7, 0.3)$ and clipping to $[0.01, 1.0]$.
  - *Severity 3:* B-factors are set to a constant value of 100.0 for all atoms; occupancies are randomized as above (no zeroing).

- *Severity 4-5:* B-factors are set to constant values of 150.0 and 200.0, respectively. In addition, a random fraction of atoms have their occupancies set to 0.0, with the zeroing fractions `zero_frac` given by `(0.4, 0.5)` for severities 4 and 5 (no zeroing at lower severities).

- **Atom Name/Element Misalignment:** This simulates common formatting errors where fixed-width columns are misaligned, leading to parsing failures (Berman et al., 2000). For a fraction `frac` of `ATOM`/`HETATM` records, we randomly apply one of two modifications: (1) the atom name (columns 13-16) is shifted one character to the left or right, or (2) the element symbol (columns 77-78) is replaced with an incorrect but common element (e.g., 'C', 'O', 'N'). The fraction `frac` is: `(0.02, 0.05, 0.10, 0.15, 0.25)`.

- **Residue Name and Numbering Anomalies:** This introduces inconsistencies in residue naming and numbering (Kleywegt & Jones, 1996). A fraction `frac_name` of residues are renamed to a chemically similar but incorrect type (e.g., `THR` to `SER`, `ILE` to `LEU`). Separately, a fraction `frac_num` of residues are assigned an insertion code (e.g., 'A') or have their residue number duplicated from an adjacent residue, creating numbering conflicts. The fractions are:

  - `frac_name`: `(0.02, 0.04, 0.07, 0.10, 0.15)`
  - `frac_num`: `(0.01, 0.02, 0.04, 0.06, 0.08)`

- **Header and Terminator Record Corruption:** This simulates truncated or improperly formatted files (Cock et al., 2009). We remove all `TER` (chain terminator) and `END` (file terminator) records. Additionally, a fraction `drop_remark_frac` of `REMARK` lines are removed, and the `HEADER` line is replaced with a corrupted placeholder. The `drop_remark_frac` is: `(0.2, 0.4, 0.6, 0.8, 1.0)`.

- **CONECT Record Loss:** This removes `CONECT` records, which explicitly define covalent bonds for ligands, cofactors, and non-standard linkages (Feng et al., 2004). Their absence forces models to infer connectivity, which can be error-prone. We randomly discard `CONECT` records, retaining only a fraction `keep_frac`: `(0.5, 0.35, 0.2, 0.1, 0.0)`. At severity 5, all `CONECT` records are removed.

## C.2 PROTEIN PERTURBATION SIMILARITY

To quantitatively assess the magnitude of structural changes induced by the geometric and coordinate-level perturbations detailed in Section C.1.1, we employ two widely accepted metrics that capture different aspects of structural similarity. Together, Root-Mean-Square Deviation (RMSD) and Template-Modeling score (TM-score) provide a complementary view of structural dissimilarity, capturing both fine-grained coordinate deviations and global topological changes, respectively.

- **Root-Mean-Square Deviation (RMSD):** This metric measures the average distance between corresponding atoms after an optimal rigid-body superposition of the two structures (Kabsch, 1976). It is highly sensitive to local coordinate deviations and serves as a gold standard for comparing highly similar conformations. A lower RMSD value indicates greater similarity. In this study, we compute the $C\alpha$-RMSD, focusing on the backbone trace of the protein. This provides a consistent measure of fold deviation, even when sidechain atoms are perturbed or deleted (as described in Section C.1.1), and is less susceptible to noise from flexible sidechain movements.

- **Template-Modeling score (TM-score):** This metric assesses the topological similarity of protein folds and is designed to be independent of protein length (Zhang & Skolnick, 2004). It produces a normalized score between 0 and 1, where a score greater than 0.5 generally indicates that two proteins share the same fold, and a score of 1.0 indicates a perfect match. Unlike RMSD, which can be heavily skewed by local deviations or flexible loops, TM-score places greater weight on the global fold similarity. This makes it particularly well-suited for evaluating perturbations that may preserve the overall topology while introducing significant local changes, such as residue deletions or geometric distortions.

### C.3 PERTURBATIONS FOR CRYO-EM IMAGES (MRC FORMAT)

Our cryo-EM perturbations target 2D particle images and are designed to simulate a range of experimental artifacts and worst-case adversarial scenarios.

#### C.3.1 IMAGE CORRUPTIONS

These corruptions mimic noise and degradation commonly found in raw cryo-EM micrographs.

- **Gaussian Noise:** To tightly couple our perturbation to the biology and the cryo-EM data-curation pipeline, we model residual detector readout/gain fluctuations after normalization as additive zero-mean Gaussian noise (McMullan et al., 2016). We apply additive noise sampled from $\mathcal{N}(0, c)$, where `c` is the standard deviation of the noise applied to the normalized image. The parameter `c` is: (0.005, 0.03, 0.05, 0.10, 0.20).

- **Shot Noise:** Because low-dose single-electron counting yields quantum arrival statistics that dominate the acquisition noise, we treat the signal fluctuations as shot (Poisson) noise and simulate them via Poisson sampling (Li et al., 2013). We model this by scaling the normalized image intensity by a factor `c`, applying a Poisson sampling process, and then rescaling. A smaller `c` corresponds to a lower signal-to-noise ratio. The parameter `c` is: (2000, 800, 300, 60, 25).

- **Speckle Noise:** Heterogeneity in vitreous-ice thickness, contamination, and illumination introduces multiplicative intensity modulations across micrographs—crucial in curation—so we apply a speckle-type multiplicative noise to mimic these field-dependent variations (Rice et al., 2018). This is modeled as $I' = I + I \cdot \mathcal{N}(0, c)$, where $I$ is the normalized image. The parameter `c` is: (0.005, 0.015, 0.03, 0.05, 0.10).

- **Gaussian Blur:** High-frequency attenuation from the CTF envelope, defocus mis-settings, and residual motion blur motivate approximating these resolution-loss mechanisms with Gaussian blurring (Zhang, 2016). We apply a Gaussian filter with a standard deviation `sigma`. The parameter `sigma` is: (0.07, 0.10, 0.15, 1.5, 4.0).

- **Low Contrast:** As unstained biomolecules in vitreous ice behave as weak-phase objects recorded under stringent low dose, we explicitly reduce image contrast to emulate the inherently low-contrast regime encountered in real datasets (Glaeser, 2013). We reduce contrast by linearly interpolating the image towards its mean value. The interpolation factor `c` ranges from 1.0 (no change) to 0.0 (zero contrast). The parameter `c` is: (0.9, 0.7, 0.5, 0.3, 0.1).

- **Impulse (Salt-and-Pepper) Noise:** Sparse extreme-valued pixels arising from hot/bad pixels, occasional cosmic-ray/electron strikes, or imperfect gain/dark normalization in DED cameras are modeled by impulse (salt-and-pepper) noise to reflect anomalies that curators routinely mask (Afanasyev et al., 2015). For each pixel, with probability $c/2$ it is set to the minimum intensity and with probability $c/2$ it is set to the maximum intensity (otherwise it is left unchanged). The parameter `c` is: (0.0005, 0.001, 0.0035, 0.01, 0.03).

- **Elastic Transform:** Beam-induced motion and specimen charging non-rigidly deform the ice film and particles, so we apply smooth elastic warps to approximate these local distortions observed during acquisition (Zheng et al., 2017). We apply a random displacement field to the image pixels, where the field is generated by filtering random noise with a Gaussian kernel. The transformation is controlled by `alpha` (scaling of displacement) and `sigma` (smoothness of displacement). The ranges for (`alpha`, `sigma`) increase with severity.

- **Translation & Rotation:** To reflect pose-estimation errors and stage/sample drift in SPA alignment/curation workflows, we inject random in-plane translations and rotations—the primary rigid parameters optimized by standard refinement packages (Scheres, 2012). We apply random 2D rotations and translations. Translations are performed efficiently in the Fourier domain, while rotations use an affine transform. Both operations leverage GPU acceleration via PyTorch. The magnitude of the transformations increases with severity, with rotation angles up to $30°$ and translations up to 25 pixels at the highest level.

### C.3.2 ADVERSARIAL PERTURBATIONS

To assess worst-case vulnerability, we employ a standard Projected Gradient Descent (PGD) attack. This is not a naturally occurring corruption but a method to find a minimal perturbation that maximally degrades model performance.

- **Projected Gradient Descent (PGD) Attack:** This iterative method generates an adversarial perturbation $\delta$ that is constrained within an $\ell_\infty$-norm ball of radius $\epsilon$. The perturbation is optimized to maximize a given loss function $\mathcal{L}$ (e.g., cross-entropy for classification tasks). The update rule at each step $t$ is:

$$x^{t+1} = \Pi_\epsilon \left( x^t + \alpha \cdot \text{sign}(\nabla_x \mathcal{L}(\theta, x, y)) \right) \tag{1}$$

  where $x^t$ is the perturbed image at step $t$, $\alpha$ is the step size, $\nabla_x \mathcal{L}$ is the gradient of the loss with respect to the input, and $\Pi_\epsilon$ is the projection operator that clips the total perturbation to be within $[-\epsilon, \epsilon]$. We use standard parameters for the number of iterations, step size $\alpha$, and perturbation budget $\epsilon$ to evaluate model robustness under this adversarial setting.

### C.4 CRYO-EM RECONSTRUCTION QUALITY METRICS

To evaluate the quality and resolution of the 3D density maps generated by the reconstruction models (e.g., CryoDRGN, CryoNeRF) from original and perturbed 2D particle images, we utilize the following standard metrics. These metrics allow us to quantify the impact of perturbations on the final reconstructed volume.

- **Q-score:** The Q-score is a per-atom metric that quantifies the resolvability of an atom by measuring the correlation between the experimental cryo-EM density map and a map generated from the atomic model (Pintilie et al., 2020). It provides a value between 0 and 1, where higher values indicate better local map-to-model agreement. In our analysis, to obtain a single quality indicator for an entire protein chain, we first compute the Q-score for every atom in the chain and then report the mean of these values. This average Q-score serves as a robust measure of the overall quality of the model's fit to the reconstructed density.

- **Fourier Shell Correlation (FSC):** FSC is the standard method for estimating the resolution of a cryo-EM reconstruction (Rosenthal & Henderson, 2003). It measures the normalized cross-correlation between two 3D maps, each reconstructed independently from a random half of the particle dataset, as a function of spatial frequency. The resolution is determined as the spatial frequency at which the FSC curve drops below a specific threshold. Following the "gold-standard" convention, we report the resolution at the FSC=0.143 criterion, which provides a reliable estimate of the achievable detail in the map. A lower resolution value (in Ångströms) indicates a higher-quality reconstruction.

## D CRYO-EM RECONSTRUCTION QUALITY RESULTS

Table 5: Results of FSC across five severities and various noise methods, reconstructed by cryo-DRGN. Each value is the average over three runs.

| Severity | Elastic | Gaussian Blur | Gaussian | Impulse | Low Contrast | Rotation | Shot | Speckle | Translation |
|---|---|---|---|---|---|---|---|---|---|
| 1 | 3.502 | 3.503 | 3.502 | 3.503 | 3.502 | 3.502 | 3.504 | 3.502 | 3.502 |
| 2 | 3.502 | 3.503 | 3.502 | 3.502 | 3.503 | 3.505 | 3.503 | 3.502 | 3.509 |
| 3 | 3.504 | 3.503 | 3.505 | 3.502 | 3.503 | 3.736 | 3.504 | 3.503 | 7.205 |
| 4 | 3.501 | 4.279 | 3.511 | 3.504 | 3.502 | 4.198 | 3.509 | 3.502 | 22.992 |
| 5 | 3.502 | 8.612 | 3.535 | 3.509 | 3.503 | 4.574 | 3.518 | 3.506 | 64.663 |

Table 5 presents the evaluation results of cryoDRGN under five severity levels. Across nine corruption methods, cryoDRGN exhibits strong robustness to all noise-based perturbations but is highly sensitive to translation operations, which cause a drastic collapse in reconstruction performance as the severity level increases. We attribute such superior robustness of Cryo-EM models (e.g.,

CryoDRGN) compared to structure/sequence models (e.g., GearNet, ProNet) to three key factors: Information Aggregation, Training Objectives, and Input Continuity:

**Information Aggregation**: Cryo-EM Models: According to our task setup (see Appendix A for more details), Cryo-EM reconstruction involves inferring a 3D density from thousands of 2D particle images. Even if individual images are perturbed (e.g., Gaussian noise or blur), the reconstruction process effectively averages out zero-mean noise across the dataset. This acts as an inherent statistical "denoising" mechanism. Structure/Sequence Models: In contrast, models like GearNet or ProNet operate on a single graph or sequence instance. There is no redundancy; if the connectivity of that single input graph is perturbed (e.g., via the radius changes shown in Figure 4), the message-passing path is fundamentally altered, leading to immediate performance degradation.

**Discrete vs. Continuous Manifolds**: Cryo-EM (Continuous): CryoDRGN operates in a continuous image/volume space using a coordinate-based neural network (VAE/MLP). Perturbations like rotation or translation result in continuous shifts in the latent space rather than discrete topological breaks, allowing the model to maintain stability. Structure Models (Graph Sensitivity): Our results in Figure 5 ("Vulnerability of Density and Spatial Modeling") reveal a mechanistic fragility in graph-based Bio-FMs. These models rely on discrete edges defined by hard cutoffs (e.g., radius or k-NN). A "tiny" ML perturbation (e.g., changing the radius from 10Å to 10.1Å) can discontinuously alter the graph topology, adding or removing edges that are crucial for message passing. This topological instability is a primary driver of the brittleness we observed.

**Inherent Data Noise and Denoising Objectives**: Cryo-EM (Low SNR Resilience): As the reviewer alludes to (and as we detail in Appendix C.3, raw Cryo-EM micrographs are inherently characterized by extremely low Signal-to-Noise Ratios (SNR) due to electron dose limitations and ice thickness. Consequently, Cryo-EM models are explicitly designed as generative denoising frameworks. During training, they are forced to learn to filter out massive amounts of stochastic noise (shot noise, background scattering) to reconstruct the underlying signal. This essentially acts as "adversarial training" by nature—the model is conditioned to be robust to noise because the noise is a dominant feature of its training distribution. Structure/Sequence (Clean Data Bias): In stark contrast, structure-based Bio-FMs (like GearNet or Inverse Folding models) are predominantly trained on PDB data, which consists of curated, solved atomic coordinates. These inputs represent a "cleaned" manifold with minimal noise. Because these models rarely encounter significant geometric noise or corruption during pre-training, they lack the learned immunity to perturbations. When we introduce "biologically plausible" noise (e.g., coordinate shifts) at inference time, it pushes the input strictly out-of-distribution for these models, leading to the fragility we observed.

## E    VISUALIZING THE ROBUSTNESS BOUNDARY UNDER BIOLOGICAL PERTURBATIONS

In Figure 9, we illustrate the relationship between input degradation and model efficacy across different Bio-FMs. By plotting the task performance against the structural dissimilarity induced by biological perturbations, we highlight the "worst-case" boundary (indicated by the lower envelope curve) to demonstrate how rapidly reliability declines even with minor input deviations.

## F    THE USE OF LARGE LANGUAGE MODELS (LLMS)

For improved clarity and readability, we relied on a large language model exclusively as an editing assistant. Its function was confined to grammar correction, style refinement, and language polishing, comparable to traditional grammar-checking software or dictionaries. The model did not generate scientific content or ideas, and its use aligns with accepted norms for manuscript preparation.

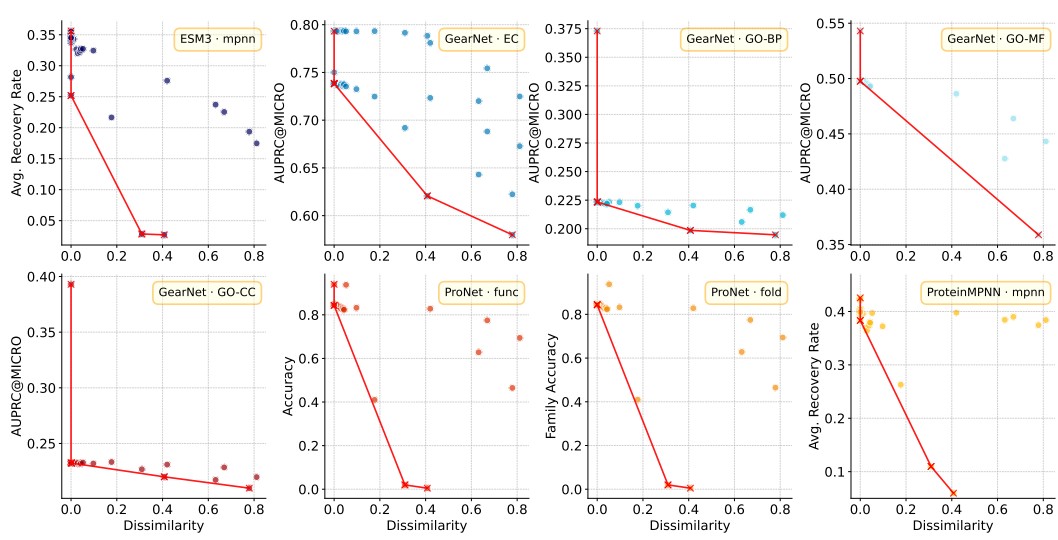

Figure 9: The robust boundary of Bio-Fms in biologically plausible perturbations.

