# OpenReview forum: "(Be Cautious!) Bio-Foundation Models Are Not Yet Robust to Biological Plausible Perturbations and ML Transformations"
_ICLR.cc/2026/Conference — ICLR 2026 Conference Withdrawn Submission_

### Official Review · Reviewer_6SoT · 2025-10-19

**Soundness:** 1
**Presentation:** 3
**Contribution:** 1
**Rating:** 0
**Confidence:** 3

**Summary:**

This paper investigates how sensitive Biological Foundation Models are to both ML and biological perturbations during inference time. In other words, their behavior out of distribution. ML perturbations are measured by varying graph radius, noise level, maximum neighbors, etc., while biological perturbations are measured by geometric distortions, Gaussian blurring, PGD, name/element scrambling, etc.

**Strengths:**

The paper is well written and the study is very thorough.

**Weaknesses:**

The paper examines two types of perturbations: biological and ML, and evaluates the sensitivity of BFMs to each.

Biological perturbations refer to data alterations such as Gaussian blurring or element scrambling.
It is unsurprising that these push the model out of distribution, as such perturbations are not part of its training regime.
When perturbations resemble the noise assumed by the model or when the algorithm (for example, CryoDRGN) includes optimization or retraining steps, the effects are naturally less severe.
These observations are well known: BFMs, like any tool, assume correct input, and using them outside those assumptions predictably leads to degraded results.

ML perturbations involve factors such as incorrect hyperparameters or altered molecular graph connectivity.
Again, these take the model out of its training distribution and predictably harm performance.
Prior work has already shown that using the wrong noise scheduler in diffusion models or applying a GNN trained on one graph family to another results in underperformance.

Overall, the paper draws no new or meaningful conclusions. It highlights expected behavior: misusing a model leads to poor results.

**Questions:**

Issue with citation on line 154-155 (Zhong)
AF3 is trained to predict protein structures mainly from X-ray data. Do the downstream tasks using these AF3 structures assume the same structural bias as X-ray determined structures, which are usually in their minimal energy state?

---

> ### Author Response · Authors · 2025-11-21
> **Official Rebuttal**
>
> We thank the reviewer for your response. However, we strongly disagree with the reviewer’s criticism and reasons for giving a strong rejection.
>
> **Bridging AI for bio and adversarial ML**. As Bio-FMs are evolving and being deployed in real scientific discovery, their robustness and reliability are largely overlooked by the community. Our work is the first comprehensive robustness study of Bio-FMs (recognized by Reviewer BAfu), and its significance and contribution are being highly commended by Reviewer BAfu (“This is the first comprehensive robustness study of Bio-FMs, addressing an important gap in current research”), azVN (“this is a primary strength of the paper in highlighting the limitations of existing FMs”; “the paper is overall strong in its content”). It is not merely testing the misuse of Bio-FMs, but highlighting and warning that those real-world feasible perturbations are being overlooked in the Bio-FM usage.
>
> Although you may believe that no system is perfectly robust and any noise will inevitably degrade performance, in the bioinformatics domain, it is crucial to identify the specific sources of perturbation and to design plausible, biologically grounded perturbations. This is precisely the goal of our work: to characterize realistic failure modes and provide a principled understanding of how Bio-FMs behave under such conditions.
>
> **A systematic study of the robustness of Bio-FMs from biological and ML perspectives**. We are targeting for systematic study aimed at helping bio experts regarding robustness issues when using current foundation models, rather than simply misusing Bio-FMs. To do that, we characterize the existing biological workflow (Figure 2) and propose perturbations from biology (Appendix C) and ML (Appendix B) perspectives. This taxonomy is neither trivial nor easy to analyze. The biological perturbations are carefully designed by biologists, and are the most common and inevitable corruptions that could happen in the real-world wet-lab. Similar ML transformations involve the preprocessing and protein structure curation process, which is complementary and systematic to the biological perspective.
>
>
> **Easy concept, but effortful studies for a non-trivial job**. Our findings are based on over 2000 experiments spanning 11 BioFMs, 7 datasets, and 4 categories of downstream tasks. We not only test the performance drop but also provide a comprehensive robustness report from different dimensions, including robust boundary (Section 4.2 Figure 3), protein density/spatial structure (Section 4.2 Figure 5), multi-modal scenarios (sequence, structure, image), and agentic pipelines. Most importantly, we offer bio interpretations, such as characterizing different Bio-FMs that show a preference to perturbation due to various encoding or tokenization, which is essential in bioinformatics.

---

> ### Author Response · Authors · 2025-11-23
> **Official Rebuttal**
>
> Moreover, we respectfully disagree that these findings are unsurprising or trivial. While it is a truism in ML that "garbage in equals garbage out," our work highlights three critical findings that challenge the current deployment of Bio-FMs:
>
> **Lack of Optimization Strategies in other Standard Bio-FMs**: The reviewer correctly notes that CryoDRGN is robust partly due to its optimization/reconstruction nature. However, this highlights precisely the critical gap we aim to expose. Unlike Cryo-EM models, the vast majority of current Bio-FMs (e.g., ESM, GearNet, ProNet) operate as static, feed-forward inference models without test-time optimization or inherent noise-adaptation strategies. They lack the "self-correction" mechanisms present in reconstruction algorithms. By benchmarking this disparity, we demonstrate that the field cannot simply rely on the architecture's implicit robustness; explicit adaptation strategies (like those used in CryoDRGN) must be integrated into sequence and structure models to ensure reliability.
>
> **Real-World "Garbage" is the Standard**: Unlike general CV/NLP domains where clean data is abundant, biological data is inherently noisy. "Biologically plausible" perturbations (e.g., thermal fluctuation, resolution limits, missing residues) represent the actual operating conditions of these models in wet-lab scenarios, not edge cases. By characterizing exactly which types of realistic noise break specific models (e.g., ProNet vs. ESM3 in Figure 8), we provide a necessary roadmap for moving these models from "benchmarks" to reliable lab tools.
>
> **The "Silent Failure" Problem**: The most concerning finding is not that performance drops, but how it drops. As shown in Figure 7 (Antibody Design), Bio-FMs often maintain high confidence (e.g., stable pLDDT or likelihood scores) even when the output is catastrophically wrong due to subtle perturbations. This decoupling of confidence and accuracy is a critical safety issue for agentic workflows where downstream tools (like Rosetta) rely on the upstream model's self-reported uncertainty.
> Imperceptible Perturbations: We demonstrate that models fail not just on "garbage," but on inputs that are scientifically valid but slightly transformed. For example, in Figure 3, we show that minimal changes to ML hyperparameters (like k-NN construction)—which are often arbitrary choices made during inference—cause massive performance distinct drops (e.g., GearNet AUPRC dropping from 0.7 to 0.1). These are not "corrupted" inputs; they are valid representations that the model should be robust to, yet isn't.
>
> > Q1: Issue with citation on line 154-155 (Zhong) AF3 is trained to predict protein structures mainly from X-ray data. Do the downstream tasks using these AF3 structures assume the same structural bias as X-ray determined structures, which are usually in their minimal energy state?
>
> The citation (Zhong et al., 2021 a;b) on lines 154-155 refers to Cryodrgn/Cryodrgn2 methodologies. We believe this is correct in the context of the sentence discussing Cryo-EM.
>
> The AF3 + Rosetta application in our paper is for screening candidates with Rosetta scoring. Though Rosetta scoring is originally tuned for experimentally determined structures (or high‐quality model conformers) where the backbone and side‐chain geometry is well‐resolved, there is increasing work where Rosetta scoring (or Rosetta refinement) is applied to predicted structures (including those from AF/AlphaFold2/3) to assess, refine, or discriminate among them [1-3].  They demonstrate that Rosetta energy discrimination can aid in analyzing point mutations and exploring conformational ensembles of AF2 models.
>
> Reference:
>
> [1] Stein, Richard A., and Hassane S. Mchaourab. "Rosetta energy analysis of AlphaFold2 models: point mutations and conformational ensembles." BioRxiv (2024): 2023-09.
>
> [2] Drake, Zachary C., Justin T. Seffernick, and Steffen Lindert. "Protein complex prediction using Rosetta, AlphaFold, and mass spectrometry covalent labeling." Nature communications 13.1 (2022): 7846.
>
> [3] Peccati, Francesca, Sara Alunno-Rufini, and Gonzalo Jiménez-Osés. "Accurate prediction of enzyme thermostabilization with Rosetta using AlphaFold ensembles." Journal of chemical information and modeling 63.3 (2023): 898-909.

---

> > ### Comment · Reviewer_6SoT · 2025-11-25
> >
> > I thank the authors for their detailed response and for the substantial experimental effort. However, my main concerns remain and my overall evaluation is unchanged.
> >
> > In my view, most of the studied perturbations amount to using the models outside the distribution and assumptions they were trained under. This includes both the so called ML perturbations such as changing graph construction or hyperparameters, and many of the biological perturbations that significantly alter the data format seen during training. From an ML standpoint, the main conclusion is that under such shifts performance can degrade and different models degrade differently. This is expected behavior.
> >
> > The comparison with methods that include an explicit reconstruction or optimization loop versus standard feed forward BioFMs also does not provide new ML insight. These are simply different problem formulations, and the fact that they respond differently under strong distribution shift is not surprising.
> >
> > Overall, the paper merely observes that under some conditions which I consider unreasonable or outside the intended usage of the models, some models perform worse than others. For these reasons, while I acknowledge the effort and the practical cautionary message, I keep my original score.

---

> ### Author Response · Authors · 2025-11-27
> **Thank you for your response**
>
> We thank the reviewer for their comments. However, we respectfully but fundamentally disagree with the premise that our work merely illustrates "unsurprising" out-of-distribution (OOD) degradation or "misuse" of models.
>
> We believe this view reflects a disconnect between **standard ML robustness** assumptions (where clean test sets are expected) and the reality of **biological deployment**, where data is inherently noisy, heterogeneous, and rarely matches the training distribution. We highlight three critical contributions that refute the claim that our conclusions are trivial:
>
> 1. **The First Bio-FM Robustness Evaluation** The reviewer suggests our perturbations are arbitrary or standard noise. We strongly clarify that we did not simply apply random Gaussian noise; we employed domain experts to design biologically plausible perturbations that mimic specific, unavoidable experimental conditions. We introduce the first comprehensive **taxonomy of perturbations**, including sidechain atom drops, disulfide bond breakage, and thermal coordinate fluctuations. Prior to our work, there was **no standard definition** or benchmark for quantifying Bio-FM robustness against wet-lab realities. By establishing this gap, we provide the community with a necessary metric to audit model reliability beyond standard leaderboard accuracy.
>
> 2. **In Biology, "Out-of-Distribution" is the Operating Condition** We strongly disagree with the claim that analyzing performance under domain shift is "meaningless" or "trivial." It is because **real-world noise is inevitable**. Unlike general CV/NLP tasks where input quality can be controlled, biological data acquisition is physically constrained and noisy (e.g., cryo-EM artifacts, low-resolution PDB regions). One can never expect real-world test data to match the idealized training distribution of models. We show that Bio-FMs are not just "less accurate" under these conditions; they exhibit catastrophic failures under conditions that are standard in experimental biology. Dismissing this as "misuse" implies that Bio-FMs are not ready for real-world scientific application. Our work quantifies exactly **how unsafe they are for their intended purpose**.
>
> 3. **Structural Fragility and "Silent Failures"** Our results reveal specific ML-side insights that go beyond simple performance drops, highlighting architectural flaws. We show that models are extremely sensitive to graph construction hyperparameters. For example, a tiny change in the interaction radius (e.g., 1%) causes performance to collapse. This reveals that current Bio-FMs rely on "hard" geometric cutoffs that create discontinuous decision boundaries, a fundamental architectural weakness that must be addressed. In terms of agentic risk & silent failure, we show that Bio-FMs often fail "silently." For instance, AlphaFold3 maintains high confidence (high ptm scores) even when generating antibodies that are physically invalid (high Rosetta energy). This is not just a performance degradation; it is a safety hazard in agentic pipelines that the reviewer overlooks.
>
> 4. **Non-FM Biological Tools are More Robust** We demonstrate that traditional non-FM tools (e.g., BLAST) are significantly more robust to these biological perturbations than modern Foundation Models (Section 5.2 Table 3), maintaining accuracy in 8 out of 12 perturbation scenarios. This highlights a regression in reliability that the ML community must actively solve, rather than accept as "expected behavior."
>
> We hope this clarifies that our work is not about applying random noise, but about rigorously stress-testing the readiness of Bio-FMs for the noisy, messy reality of biological science.

---

### Official Review · Reviewer_azVN · 2025-10-21

**Soundness:** 3
**Presentation:** 2
**Contribution:** 3
**Rating:** 6
**Confidence:** 3

**Summary:**

This paper offers a comprehensive analysis of the capabilities and limitations of a suite of biological foundation models (BioFMs) by measuring their robustness against varying degrees of perturbations to their inputs. Their primary finding -- the overall high sensitivity and lack of robustness of most BioFMs -- is a valuable contribution, especially in the context of applications of deep representation learning to the physical sciences. This finding is supported by a large amount of empirical evidence (over 2000 experiments spanning 11 BioFMs, 7 datasets, and 4 categories of downstream tasks) in the context of different kinds of perturbations of the inputs (ML perturbations and biologically plausible perturbations).

**Strengths:**

- Originality: this is a standard kind of ML paper that measures robustness (or lackthereof) of various deep learning models, and it is overall pursued in a reasonable manner.
- Quality: the primary qualitative strength of this paper is its comprehensiveness in the wide variety of experiments, tasks, models, and datasets they perform the evaluations upon, as well as the framing of two different forms of perturbations (ML vs Bio) and the lack of robustness of these models in the face of both regimes.
- Clarity: the paper is overall clear in its framing, however it would be difficult for any non-Bio reader to follow the details of their experiments and approach.
- Significance: this is a primary strength of the paper in highlighting the limitations of existing FMs, as well as its determination of which BioFMs are more or less resilient than others in handling these kinds of data drifts (e.g., Cryo-EM reconstruction models like CryoDRGN are fairly robust).

**Weaknesses:**

1. I believe that the paper is overall strong in its content, with one substantive difficulty. Namely, it exclusively evaluates BioFMs and does not compare to traditional baselines or linear models or standard tools. For example, the authors mention both in the abstract and in lines 206-207: "Tiny biological perturbations [...] may be invisible to standard tools but can catastrophically alter Bio-FM outputs". This claim, which I presume is true, is however not substantiated by evidence in the paper (as well, what are these standard tools exactly is not discussed). I believe that the work would be significantly strengthened to a clear Accept if it were possible, for each of the experiments and figures in question (or at least for a subset), to include how the performance of a standard tool or method would change in these benchmarks, under those perturbations. This would be a majorly helpful baseline and point of reference, as we (the community, readers, and the authors) would then have an improved understanding of under what perturbational conditions and at which points (i.e. at what level of perturbation severity) when BioFMs start to underperform compared to standard tools; e.g., for tasks such as function and property prediction. This would also substantiate the claim that the standard tools / non-FM methods are not in fact vulnerable to such perturbations.

1. While not a weakness per-se, but an opportunity for strength (although arguably beyond the scope of a benchmarking paper) would of course be to leverage these insights to design an improvement to the bio FM regime (or perhaps a more simple finetuning methodology) that can attempt to address these issues of robustness.

1. The other primary weakness is of secondary importance, and I believe could be addressed by the camera-ready deadline. This is the overall presentation and quality of the figures and layout, which could be substantially improved:
   - Figure 1: is it not more standard to have the abstract above the first figure?
   - Figure 1 and 2: why use comic-sans font for parts of these figures? It appears somewhat unprofessional.
   - 197-198: the use of the plural "questions" is immediately followed by only one question, consider rephrasing.
   - The section titles (3, 4, 5) are rather long, normative claims, and atypical. The rest of the paper includes claims and arguments, I am not sure if section titles should also repeat those (rather than simply being: "Assessing Robustness" / "Robustness to ML transformations" / "Robustness to Biologically Plausible Transformations"
   - Lack of figure standardization in results. Some figures are PNGs and some are PDFs with highlightable text -- typically it is preferred to have all figures be PDFs with highlightable text. Furthermore, various figures have different fonts, font sizes, and coloring standards.
   - The use of in-text Figures may not be standard formatting for ICLR, and it is overall confusing with Figure 8 being referenced before Figure 7 but both being on separate pages. I think figures should not be in-line and instead should be across the page as is done for Figures 3 and 4.
   - Appendix section E repeats the same sentence 3 times (in the section title, in the 1 sentence paragraph, and in the figure caption)

**Questions:**

- The figures can be confusing -- why does Figure 6 have different y-axis labels for the different models under question? Is that an error, or are Avg. Recovery Rate and Accuracy only relevant measures of task performance for specific kinds of models?

---

> ### Author Response · Authors · 2025-11-22
> **Official Rebuttal**
>
> > W1: I believe that the paper is overall strong in its content, with one substantive difficulty. Namely, it exclusively evaluates BioFMs and does not compare to traditional baselines or linear models or standard tools.
>
> We thank you for your insightful comment. We agree it is necessary to provide empirical evidence to support that traditional non-FM biological tools are robust to those perturbations compared to Bio-FMs. As suggested, we take enzyme function prediction as an example and use BLAST as a non-FM conventional tool. BLAST transfers the EC annotation of the closest homologous sequence identified through high-scoring alignments.
>
> We conduct the experiments on three enzyme function prediction datasets as described in [1], and apply the same 11 biological perturbation categories with severity equals to 1 and 3. Results are summarized in Table 1.
>
> **Table 1:** Effects of different perturbation methods on protein BLAST accuracy across three datasets and two severity levels.
>
> | Dataset | halogenase | halogenase |multi | multi |new | new |
> | :--- | :---: | :---: |:---: | :---: |:---: | :---: |
> | Ori BLAST | 0.66 | 0.66 | 0.18 | 0.18 | 0.52 | 0.52 |
> | **Bio. Perturb.** | **Severity1** | **Severity3** |**Severity1** | **Severity3** |**Severity1** | **Severity3** |
> | Gaussian Coordinate Noise | 0.66 | 0.59 |0.18 | 0.18 | 0.52 | 0.51 |
> | Local Residue Deletion | 0.62 | 0.62 | 0.18 | 0.15 | 0.51 | 0.50 |
> | Sidechain Atom Drop | 0.66 | 0.66 | 0.18 | 0.18 | 0.52 | 0.52 |
> | Disulfide Bond Breakage | 0.66 | 0.66 | 0.18 | 0.18 | 0.52 | 0.52 |
> | Cis-Peptide Bond Error | 0.66 | 0.66 |0.18 | 0.18 | 0.52 | 0.52 |
> | Local Geometric Distortion | 0.66 | 0.66 | 0.18 | 0.18 | 0.52 | 0.52 |
> | B-Factor and Occupancy Scrambling | 0.66 | 0.66 | 0.18 | 0.18 | 0.52 | 0.52 |
> | Atom Name/Element Misalignment | 0.66 | 0.66 | 0.18 | 0.18 | 0.52 | 0.52 |
> | Residue Name and Numbering Anomalies | 0.59 | 0.66 |  0.18 |  0.18 | 0.52 | 0.52 |
> | Header and Terminator Record Corruption | 0.66 | 0.66 | 0.18 | 0.18 | 0.52 | 0.52 |
> | CONECT Record Loss | 0.66 | 0.66 | 0.18 | 0.18 | 0.52 | 0.52 |
>
> We show that BLAST is not being affected in 8 out of 12 perturbations. This is because BLAST takes a sequence as input and matches the enzyme function via sequence similarity from external databases. Naturally, it immunizes spatial distance perturbations such as the Gaussian coordinate perturbation. While Bio-FMs models the spatial protein structures and significantly suffers from Gaussian coordinate perturbation (e.g., GearNet drops from 0.76 to 0.65). Even for the rest 4 perturbations, BLAST demonstrates strong resilience, e.g., BLAST only drops 3-4% accuracy on average and drops only 7% at most in the worst scenario.
>
> This shows that although Bio-FMs are highly capable on many tasks, they also exhibit greater vulnerability compared to traditional non-FM bio tools. We will expand our discussion and include additional results, such as broader comparisons between Bio-FMs and non-FM tools on the same tasks, to further illustrate this point in our manuscript.
>
> Reference:
> [1] Yu, Tianhao, et al. "Enzyme function prediction using contrastive learning." Science 379.6639 (2023): 1358-1363.
>
> > W2: While not a weakness per-se, but an opportunity for strength (although arguably beyond the scope of a benchmarking paper) would of course be to leverage these insights to design an improvement to the bio FM regime (or perhaps a more simple finetuning methodology) that can attempt to address these issues of robustness.
>
> We thank you for your comment. We agree that designing an improvement to the Bio-FM is essential. Our work will serve as the necessary precursor, providing a comprehensive profiling of Bio-FM robustness behaviors. We will take the robustness enhancement as future research.
>
> > W3: The other primary weakness is of secondary importance, and I believe could be addressed by the camera-ready deadline.
>
> Thank you very much for your careful proofreading. We will update our manuscript according to your suggestions.
>
> > Q1: The figures can be confusing -- why does Figure 6 have different y-axis labels for the different models under question? Is that an error, or are Avg. Recovery Rate and Accuracy only relevant measures of task performance for specific kinds of models?
>
> Sorry for the confusion. In Figure 6, different metrics mean different tasks. Avg. Recovery Rate is for the inverse folding task, and Accuracy is for protein function prediction. We will annotate the corresponding task and dataset in each subfigure.

---

> ### Comment · Reviewer_azVN · 2025-11-27
>
> Thank you for your answers and for the addition of these results. This table includes the results for BLAST on this enzyme function prediction task (a seemingly new task not evaluated in other parts of this work). It is unclear to me how to compare these results to those that would be obtained by a BioFM on this task; you say "GearNet drops from 0.76 to 0.65" on "Gaussian coordinate perturbation" for this task, but my understanding is that this task is different from the others presented in this work. Therefore, for clarity, completeness, and for the sake of apples-to-apples comparisons, I think it would make sense to include GearNet on each metric in this table. For example, the formatting could be something like:
>
> ```
> BLAST / GearNet | halogenase | ...
> ----------------------------------------
> Unperturbed        | 0.66 / 0.76   | ...
> *Bio perturb.*    | sev 1 | sev3 | sev1 | sev3 ...
> Gaussian Coordinate Noise	|  0.66 / 0.65 | ...
> ```
>
> And then the readers and potential users of these tools could clearly see exactly under what kind of perturbational conditions it would be preferable to use this non-FM tool rather than using GearNet.

---

> > ### Author Response · Authors · 2025-12-03
> >
> > We thank you for your insightful comment. As suggested, we conduct BLAST on the same dataset, EC, as GearNet and directly compare their performance under the same perturbations:
> >
> > | Dataset | BLAST (1) | GearNet (1) | BLAST (3) | GearNet (3) |
> > | :--- | :---: | :---: | :---: | :---: |
> > | Original | 0.7620 | 0.795 | 0.7620 | 0.795
> > | Gaussian Coordinate Noise | 0.7620 | **0.735** | 0.7569 | **0.724**
> > | Local Residue Deletion  | 0.7634 | **0.723**| 0.7620 | **0.643**
> > | Sidechain Atom Drop  | 0.7620 | **0.738** | 0.7620 | **0.738**
> > | Disulfide Bond Breakage  | 0.7620 | 0.793 | 0.7620 | 0.793
> > | Cis-Peptide Bond Error  | 0.7620 | **0.736** | 0.7620 | **0.735**
> > | Local Geometric Distortion  | 0.7620 | 0.793 | 0.7620 | 0.793 |
> > | Header and Terminator Record Corruption  | 0.7620 | 0.793 | 0.7620 | 0.793
> >
> > Here, BLAST (1) means perturb severity equals 1 and so on. It is shown that BLAST is more robust than GearNet, with most of the perturbations not significantly affecting BLAST performance, while GearNet is significantly affected.

---

### Official Review · Reviewer_NyP6 · 2025-10-28

**Soundness:** 2
**Presentation:** 3
**Contribution:** 2
**Rating:** 4
**Confidence:** 3

**Summary:**

This work focuses on analyzing the robustness of Bio-FMs from both biology and machine learning (ML) perspectives. In the paper, the authors systematically evaluate various state-of-the-art Bio-FMs on a spectrum of protein-related downstream tasks, spanning protein design, generation, function prediction, cryo-EM reconstruction, and structure classification, over structure, sequence, and image modalities. The results reveal that most Bio-FMs are vulnerable to both ML transformations and biological perturbations. Even though some of the perturbations are not observable by the biological measuring tools, they can still affect the outputs of Bio-FMs.

**Strengths:**

1, The challenges and motivation behind this paper are clearly shown and justified.

2. Extensive experiments on various state-of-the-art models and downstream tasks provide sufficient analysis to support the findings on the vulnerability of the Bio-FMs.

3. The paper is well written and organized.

**Weaknesses:**

1. This paper mainly focuses on the findings that Bio-FMs are vulnerable to biological and machine learning perturbations. However, as machine learning models are trained by data, the inherent limitations of these models are vulnerable to input data and process perturbations. Then what are the methods to monitor and prevent the ineffectiveness of Bio-FMs that should also be included in the paper to provide a more complete analysis? Based on my understanding,  methods such as uncertainty quantification have already been proposed to address these issues. Will these methods work on Bio-FMs?

2. When analyzing the machine learning transformation perturbations, the authors conduct the experiments by changing the parameters of the models. This is not sufficient to show that Bio-FMs are vulnerable to perturbations, as the models may become non-optimal when changing the parameters. And the degradation of models' performance is also reasonable under these circumstances.

**Questions:**

1. In section 4.2, why use the parameter k in a KNN when constructing the multi-relational GNN as the perturbations? Based on my understanding, these are the model parameters. When you change them, you change the model, and of course, non-optimal parameters will lead to the degradation of model performance.

2. Same as the previous question. Changing the parameters in the Bio-FMs is not a good perturbation method. This shows that the models' performance is sensitive to the parameters. But when the optimal parameters are given, whether the model is still sensitive to various protein structures is what is more important.

---

> ### Author Response · Authors · 2025-11-21
> **Official Rebuttal**
>
> >W1: What are the methods to monitor and prevent the ineffectiveness of Bio-FMs that should also be included in the paper to provide a more complete analysis
>
> Thank you for raising this important point. We agree that monitoring and preventing model failures is a critical direction for deploying Bio-FMs responsibly (as also mentioned by reviewer *azVN*, though not a weakness). However, our paper focuses on a necessary precursor to those methods: characterizing the failure modes and understanding the robustness landscape of Bio-FMs.
>
> At present, there is limited evidence showing whether existing reliability tools are effective for large Bio-FMs. Without first identifying where Bio-FMs fail and how vulnerable, it is difficult to evaluate whether such monitoring/prevention methods are appropriate, sufficient, or require domain-specific adaptations. Our work therefore serves as a foundational diagnostic step: we systematically reveal the types of biological and ML perturbations that degrade Bio-FM performance, quantify their impact, and analyze which structural aspects of proteins trigger vulnerabilities. This understanding is a prerequisite for designing or adapting mitigation strategies.
>
> In summary, we view robustness characterization as Step 1 toward reliability-aware Bio-FM pipelines. For potential monitoring and prevention methods, we will study the training receipts for improved robustness (such as data augmentation, biological noisy training, etc)  and make robust prediction with randomized smoothing and uncertainty quantifications in the future.
>
>
> W2&Q1&Q2: Why change the default values of model parameters? It will diverge from the optimal settings and result in poor performance.
>
> Thank you for your insightful comment. First of all, the optimal parameters are only defined for the in-domain test distribution, whereas Bio-FMs are typically deployed in open-set biological environments where the true optimal parameters are unknown. Real-world biomolecular data are also far from perfectly optimal: proteins may be modeled at different resolutions, processed through heterogeneous pipelines, or contain intrinsic structural uncertainty. Our parameter perturbation experiments therefore simulate these non-ideal, worst-case in-domain shifts to reveal how Bio-FMs respond when confronted with unknown or noisy test-domain conditions. Sensitivity to such structural variations reflects genuine vulnerability in practical deployments, not merely sensitivity to model hyperparameters.
>
> Moreover, we would like to highlight that these hyperparameters are **structural parameters that determine how protein residue neighborhoods are constructed before being fed into the Bio-FM**. Changing them is therefore equivalent to altering the spatial relationships among residues, i.e., perturbing the protein’s structural input graph, mimicking some biological perturbation scenarios. For example, reducing K from k to k−1 with a fixed cutoff radius \tau can be interpreted as constructing a less dense local neighborhood around each residue. Biologically, this corresponds to slightly increasing certain inter-residue distances so that they fall outside the cutoff and are no longer considered neighbors. Thus, modifying these “parameters” is simply a convenient ML formulation of perturbing the protein structure itself, conceptually similar to coordinate perturbation in biological perturbations, but complementary and systematic.
> In this sense, the perturbations we apply are input-level structural perturbations. They therefore reveal how sensitive a Bio-FM is to small, plausible variations in its structural inputs. For instance, we observe that GearNet is extremely sensitive to those structural variations (Figure 4 (1)), yet ESM-GearNet (Figure 4 (2)) and ProteinMPNN (Figure 4 (3)) are rather robust to those variations.

---

> ### Comment · Reviewer_NyP6 · 2025-11-25
>
> Thanks for the rebuttal. I'd like to maintain my score.

---

### Official Review · Reviewer_BAfu · 2025-11-01

**Soundness:** 3
**Presentation:** 3
**Contribution:** 3
**Rating:** 6
**Confidence:** 4

**Summary:**

This paper presents the first systematic study of robustness in Biological Foundation Models (Bio-FMs) - a class of large models trained on biological data for applications such as protein design, molecular analysis, and drug discovery. The authors approach robustness from two complementary perspectives: (1) Biological data curation -  introducing biologically plausible perturbations (e.g., noise in structure coordinates, folding errors, experimental corruption) that mimic real-world biological data noise. (2) Machine learning transformations - examining ML-specific perturbations such as changes in data preprocessing, feature embeddings, and model hyperparameters (e.g., graph construction parameters like kNN radius).

Across 2,128 experiments on 11 state-of-the-art Bio-FMs, covering 7 datasets and 4 major downstream task types (e.g., protein design, function prediction, cryo-EM reconstruction), the study finds that most Bio-FMs are not robust to even minor perturbations. Cryo-EM reconstruction models (like CryoDRGN) show notable robustness, maintaining stability even under adversarial attacks. Subtle biological perturbations, often undetectable by current tools, can cause critical failures in predictions. The authors highlight the urgent need for robustness benchmarks and provide a conceptual framework to evaluate the trustworthiness of Bio-FMs.

**Strengths:**

+ This is the first comprehensive robustness study of Bio-FMs, addressing an important gap in current research - trustworthiness in biological modeling.

+ The paper establishes a clear two-dimensional framework (biological vs. ML robustness), bridging disciplinary perspectives in biology and machine learning.

+ The inclusion of multiple modalities (sequence, structure, image) and a large number of models and datasets enhances generalizability and credibility.

**Weaknesses:**

- While the paper identifies robustness differences (e.g., CryoDRGN’s stability), it provides limited mechanistic insight into why certain architectures or representations yield better robustness.

- The robustness analysis remains entirely in silico. Validation through wet-lab experiments or real-world deployment tests would enhance credibility and biological relevance.

**Questions:**

Why do Cryo-EM models (e.g., CryoDRGN) exhibit higher robustness compared to sequence or structure based Bio-FMs? Is it due to architecture, data modality, or inherent signal redundancy in cryo-EM images?

Are the biological perturbations calibrated to realistic experimental noise levels or adversarial magnitudes?

---

> ### Author Response · Authors · 2025-11-21
> **Official Rebuttal**
>
> > W1&Q1. Why do Cryo-EM models (e.g., CryoDRGN) exhibit higher robustness? While the paper identifies robustness differences (e.g., CryoDRGN’s stability), it provides limited mechanistic insight into why certain architectures or representations yield better robustness.
>
> Thank you for your insightful comment. Based on our benchmark results (Table 2 and Figure 5) and architectural analysis, we attribute the superior robustness of Cryo-EM models (e.g., CryoDRGN) compared to structure/sequence models (e.g., GearNet, ProNet) to three key factors: Information Aggregation, Training Objectives, and Input Continuity.
>
> 1. **Information Aggregation**: Cryo-EM Models: As noted in our task setup (Section A.1), Cryo-EM reconstruction involves inferring a 3D density from thousands of 2D particle images. Even if individual images are perturbed (e.g., Gaussian noise or blur), the reconstruction process effectively averages out zero-mean noise across the dataset. This acts as an inherent statistical "denoising" mechanism.
> Structure/Sequence Models: In contrast, models like GearNet or ProNet operate on a single graph or sequence instance. There is no redundancy; if the connectivity of that single input graph is perturbed (e.g., via the radius changes shown in Figure 4), the message-passing path is fundamentally altered, leading to immediate performance degradation.
>
> 2. **Discrete vs. Continuous Manifolds**: Cryo-EM (Continuous): CryoDRGN operates in a continuous image/volume space using a coordinate-based neural network (VAE/MLP). Perturbations like rotation or translation result in continuous shifts in the latent space rather than discrete topological breaks, allowing the model to maintain stability.
> Structure Models (Graph Sensitivity): Our results in Figure 5 ("Vulnerability of Density and Spatial Modeling") reveal a mechanistic fragility in graph-based Bio-FMs. These models rely on discrete edges defined by hard cutoffs (e.g., radius or k-NN). A "tiny" ML perturbation (e.g., changing the radius from 10Å to 10.1Å) can discontinuously alter the graph topology, adding or removing edges that are crucial for message passing. This topological instability is a primary driver of the brittleness we observed.
>
> 3. **Inherent Data Noise and Denoising Objectives**: Cryo-EM (Low SNR Resilience): As the reviewer alludes to (and as we detail in Appendix C.3), raw Cryo-EM micrographs are inherently characterized by extremely low Signal-to-Noise Ratios (SNR) due to electron dose limitations and ice thickness. Consequently, Cryo-EM models are explicitly designed as generative denoising frameworks. During training, they are forced to learn to filter out massive amounts of stochastic noise (shot noise, background scattering) to reconstruct the underlying signal. This essentially acts as "adversarial training" by nature—the model is conditioned to be robust to noise because the noise is a dominant feature of its training distribution.
> Structure/Sequence (Clean Data Bias): In stark contrast, structure-based Bio-FMs (like GearNet or Inverse Folding models) are predominantly trained on PDB data, which consists of curated, solved atomic coordinates. These inputs represent a "cleaned" manifold with minimal noise. Because these models rarely encounter significant geometric noise or corruption during pre-training, they lack the learned immunity to perturbations. When we introduce "biologically plausible" noise (e.g., coordinate shifts) at inference time, it pushes the input strictly out-of-distribution for these models, leading to the fragility we observed.

---

> ### Author Response · Authors · 2025-11-21
> **Official Rebuttal**
>
> W2. The robustness analysis remains entirely in silico. Validation through wet-lab experiments or real-world deployment tests would enhance credibility and biological relevance.
>
> We thank the reviewer for this insightful comment. We agree that wet-lab validation is the ultimate benchmark for a model's predictive accuracy. However, the goal of our robustness analysis is distinct: it assesses the model's robustness under perturbation, not the absolute biological fidelity of its outputs. Therefore, wet-lab validation of perturbed prediction is out of the research scope.
>
> Bio-FMs are used to computationally generate, rank, and select a small number of high-quality candidates (e.g., protein sequences, structures) from a massive search space for subsequent, costly wet-lab validation. Our robustness analysis evaluates whether various perturbations can significantly corrupt the Bio-FM's ability to perform this digital pre-screening.
> - If the Bio-FM's output scores (like AF3 confidence or prediction scores) remain stable despite a perturbation, the set of top-ranked candidates selected for wet-lab remains virtually unchanged. Therefore, the chance of discovering a high-quality final product is preserved, and the wet-lab effort is utilized effectively.
> - If the perturbation significantly degrades the model's scores, the set of top candidates shifts or lowers in quality. This directly leads to a reduction in the success rate during the subsequent wet-lab phase, resulting in wasted human effort, time, and resources, the very outcome Bio-FMs are intended to prevent.
>
> Our robustness perspective focuses on verifying the integrity of the digital selection process. Since a demonstrated vulnerability in the Bio-FM already logically implies a high cost to wet-lab efforts (due to poor candidate prioritization), the in-silico analysis is necessary and sufficient to verify the model's resilience in its defined role.
>
> >Q2: Are the biological perturbations calibrated to realistic experimental noise levels or adversarial magnitudes?
>
> Our biological perturbations and their strength (or noise levels) are carefully selected by biologists with years of experience in protein structure analysis. These perturbations are well-recognized to be common and inevitable in literature. In Appendix C 1.1 and C 1.2, we provide the detailed literature support of our perturbations.

---

> > ### Comment · Reviewer_BAfu · 2025-11-27
> > **Keeping an optimistic borderline-positive score despite remaining concerns**
> >
> > "Our biological perturbations and their strength (or noise levels) are carefully selected by biologists with years of experience in protein structure analysis."
> >
> > The reviewer has limited biology background. In ML/AI/CS, we are used to more rigorous arguments (theoretical or empirical).  Not all concerns of the reviewer have been met but I am inclined to retain an optimistic borderline-positive score assuming that the other reviewers can better examine the biological plausibility (which is a key element of the paper and authors have ruled out any wet-lab experimentation as out of scope).
> >
> > Overall, the paper provides very limited contribution to ML/AI and it might be a better fit for a bio venue focused on applications.

---

### Note · Authors · 2026-01-28

I have read and agree with the venue's withdrawal policy on behalf of myself and my co-authors.

---

### Meta-Review · Area_Chair_NHs5 · 2026-01-07

**Summary:**

The paper proposes a synthetic study of robustness in "Biological Foundation Models" (large models trained on biological data) for different applications.  The evaluation is on biological data curation and machine learning transformations.

The paper received mixed reviews, but there is a trend on them that the paper presents known issues to the ML community to a bio audience.  While there is value on the particular domain, the contribution to the ML community is limited given the lack of insights or modifications to the models.  In this way, the paper may be better received in a bio venue instead.  Thus, I recommend the rejection of this paper.

Strengths:
- First comprehensive robustness study for "biological foundation models"
- The challenges and motivation for these models is shown and justified
- Bridges disciplinary perspectives in biology and machine learning
- The inclusion of multiple biological modalities enhances generalization and credibility
- Comprehensive and varied experiments

Weaknesses:
- The paper identifies robustness differences with limited mechanistic insights
- The robustness analysis is limited to in silico
- There are no methods to monitor and prevent the ineffectiveness of the foundation models (no designs to improve the models)
- The degradation of the models is expected, and it is not sufficient to show that the models are vulnerable to perturbations
- Lack of comparisons against traditional baselines or linear models

**Reviewer Concerns:**

Reviewer BAfu raised concerns about the limited mechanistic insights of the robustness, and that the validation is only in silico without wet-lab or real world evaluation.  The authors expanded in the rebuttal about the factors that make Cryo-EM models superior.  Regarding the limited in silico experiments, the authors mentioned that the goal of the study is to assess the model's robustness under perturbations to later be studied in costly wet-lab validation.

Reviewer NyP6 raised concerns with the methods to overcome the limitations of the foundation models, and commented on the limited contributions to overcome the limitations of the models to the perturbations.

Reviewer azVN raised concerns about the limited comparisons against standard baselines and the lack of insights on how to improve the proposed foundation models.  Moreover, the reviewer raised several minor concerns regarding the presentation of the work.  The authors replied to the reviewer's concerns and provided additional results that show that the methods are sensitive to perturbations.

Reviewer 6SoT raised significant concerns regarding the motivation behind the proposal.  It is known that misusing a model leads to poor results and that using a model under different assumptions will lead to poor results.  The authors highlight in the rebuttal that the objective of the paper is to warn the bio audience about the known limitations in ML.

**Reviewer Scores:**

Reviewer BAfu recommened a weak accept.  The authors replied to the reviewer's concerns.  While the reviewer is optimistic, they raised further concerns about the limited contribution to ML and suggests that the paper may be better suited for a bio venue.  Moreover, the reviewer commented that not all the concerns were resolved.

Reviewer NyP6 recommended a weak reject. The authors argued in the rebuttal that the study and understanding of the limitation of these method is of major importance to the bio audience that are not aware of these limitations.  The reviewer maitained their score after the rebuttal.

Reviewer azVN recommended a weak accept.  The authors provided additional results and answers to the concerns.  But there were still unaddressed issues regarding how to improve the foundation models based on the degradation results.

Reviewer 6SoT recommended a strong reject.  While the authors provided a rebuttal, it mainly focuses on showing to the bio audience the known limitations of ML models.  The reviewer maintained that the results are expected.

---

### Decision · Program_Chairs · 2026-01-26

Reject